# Genes in human obesity loci are causal obesity genes in *C. elegans*

**Wenfan Ke**[1], **Jordan N. Reed**[2], **Chenyu Yang**[1], **Noel Higgason**[1], **Leila Rayyan**[1], **Carolina Wählby**[3], **Anne E. Carpenter**[4], **Mete Civelek**[2,5], **Eyleen J. O'Rourke**[1,6]*

1 Department of Biology, College of Arts and Sciences, University of Virginia, Charlottesville, Virginia, United States of America, 2 Department of Biomedical Engineering, School of Engineering and Applied Science, University of Virginia, Charlottesville, Virginia, United States of America, 3 Department of Information Technology and SciLifeLab, Uppsala University, Uppsala, Sweden, 4 Imaging Platform, Broad Institute of MIT and Harvard, Cambridge, Massachusetts, United States of America, 5 Center for Public Health Genomics, School of Medicine, University of Virginia, Charlottesville, Virginia, United States of America, 6 Department of Cell Biology, School of Medicine, University of Virginia, Charlottesville, Virginia, United States of America

* ejorourke@virginia.edu

**Data Availability Statement:** All relevant data are within the manuscript and its Supporting Information files. Figures that have associated Raw data include: Figs 1E, 2B, 2C, 2F, 3A, 3B, 3C, 4B, 4C, 4E, 4F, 4G, 4H, 4I 4J, 4K, 4L, 5B, 5C, 5D, 5E,

## Abstract

Obesity and its associated metabolic syndrome are a leading cause of morbidity and mortality. Given the disease's heavy burden on patients and the healthcare system, there has been increased interest in identifying pharmacological targets for the treatment and prevention of obesity. Towards this end, genome-wide association studies (GWAS) have identified hundreds of human genetic variants associated with obesity. The next challenge is to experimentally define which of these variants are causally linked to obesity, and could therefore become targets for the treatment or prevention of obesity. Here we employ high-throughput *in vivo* RNAi screening to test for causality 293 *C. elegans* orthologs of human obesity-candidate genes reported in GWAS. We RNAi screened these 293 genes in *C. elegans* subject to two different feeding regimens: (1) regular diet, and (2) high-fructose diet, which we developed and present here as an invertebrate model of diet-induced obesity (DIO). We report 14 genes that promote obesity and 3 genes that prevent DIO when silenced in *C. elegans*. Further, we show that knock-down of the 3 DIO genes not only prevents excessive fat accumulation in primary and ectopic fat depots but also improves the health and extends the lifespan of *C. elegans* overconsuming fructose. Importantly, the direction of the association between expression variants in these loci and obesity in mice and humans matches the phenotypic outcome of the loss-of-function of the *C. elegans* ortholog genes, supporting the notion that some of these genes would be causally linked to obesity across phylogeny. Therefore, in addition to defining causality for several genes so far merely correlated with obesity, this study demonstrates the value of model systems compatible with *in vivo* high-throughput genetic screening to causally link GWAS gene candidates to human diseases.

6B, 6C, 6D, 7A, 7B, 7C, 7D, 7E and S1E, S3A, S3B, S4A, S4B, S4C, S4D, S4E Figs. Raw images are available from the corresponding author upon request. All reagents presented in this study and necessary to replicate the findings are available from the corresponding author upon request.

**Funding:** M.C is funded by National Institutes of Health (DK118287) A.E.C. is funded by National Institutes of Health (GM122547) E.J.O'R. is funded by Pew Charitable Trusts (Biomedical Scholars Award), National Institutes of Health (DK087928), W. M. Keck Foundation, and Jeffress Trust Award The funders had no role in study design, data collection and analysis, decision to publish, or preparation of the manuscript. NIH (DK118287) supported the salaries of M.C. and J.N.R. NIH (T32 HL007284) supported the salary of J.N.R. NIH (GM122547) supported the salary of A.E.C. All other funders supported salaries of E.J.O'R., W.K., L.R.

**Competing interests:** The authors have declared that no competing interests exist.

## Author summary

Human GWAS have identified hundreds of genetic variants associated with human obesity. The genes being regulated by these variants at the protein or expression level represent potential anti-obesity targets. However, for the vast majority of these genes, it is unclear whether they cause obesity or are coincidentally associated with the disease. Here we use a high-throughput genetic screening strategy to test *in vivo* in *Caenorhabditis elegans* the potential causal role of human-obesity GWAS hits. Further, we combined the results of the genetic screen with analyses of mouse and human GWAS databases. As a result, we present 17 genes that promote or prevent *C. elegans* obesity, and the early onset of organismal deterioration and death associated with obesity. Further, the sign of the correlation between the expression levels of the human genes and their associated clinical traits matches, for the most part, the phenotypic effects of knocking down these genes in *C. elegans*, suggesting conserved causality and pharmacological potential for these genes.

## Introduction

Obesity is a major risk factor for serious comorbidities including cardiovascular disease (CVD), type 2 diabetes, hypertension, stroke, neurodegenerative disease, and certain cancers [1,2]. In the past decade, obesity-derived comorbidities caused more than 4 million deaths per year and cost an average of 4 years of life lost worldwide [3–5]. The increased prevalence of obesity and extreme obesity could lead to a global average reduction in life expectancy in the next decade [6,7]. Therefore, there is an urgent need to improve the toolset to reduce the burden of obesity [8–10]. Identifying more effective points of intervention to treat obesity is difficult because obesity is a complex disease influenced by various interacting factors including socioeconomic status, physical activity, eating habits, microbiota, and multiple genes acting in multiple tissues [11].

On one side, the rapid increase in the prevalence of obesity observed in recent decades has been attributed to an "obesogenic environment, which offers ready access to high-calorie foods but limits opportunities for physical activity" (www.CDC.gov). Studies aimed to elucidate the role of carbohydrates in obesity suggested that high sugar consumption could be the main factor behind the alarming increase in the incidence of obesity and metabolic syndrome. More specifically, increased fructose intake from high-fructose corn syrup was found to be highly correlated with the increase in the incidence of obesity observed in the past 40 years [12–14]. Compared to glucose, fructose consumption is more effective at promoting excessive visceral fat accumulation [15] [16]. In line with this, in both humans and rodents, high-fructose diets (HFrD) are functionally linked to the general metabolic dysregulation associated with obesity, also known as metabolic syndrome [17–20].

On the other side, even when we share an obesogenic environment not everyone becomes obese, implying that a sizable portion of the variation in weight among adults would be due to genetic factors. Further, protein-coding genes whose sole inactivation is sufficient to prevent, ameliorate, or revert obesity equate to promising druggable targets. Genome-wide association studies (GWAS) are one of the most promising approaches to identify these gene targets because GWAS search for genetic variants associated with obesity in human populations living in their daily environments (as opposed to animal models reared in controlled conditions). Once identified, the genetic variants can be mapped to genes that may increase or decrease the likelihood of obesity occurrence [21]. Currently, there are over 90 GWAS, which associate 2,537 SNPs with different obesity metrics including body mass index (BMI), waist to hip ratio (WHR), WHR adjusted for BMI, body fat distribution, and/or body fat percentage [22].

Almost 90% of these variants are found in non-coding regions of the genome [23]; hence, very few variants have been mapped to genes. Further, even for those variants that can be reliably mapped to genes, after two decades, only a handful have been causally linked to the disease [24–34]. A major barrier to distinguish statistical association from causal association for the hundreds of loci associated with obesity or diet-induced obesity (DIO) is the lack of *in vivo* systems that enable experimental testing at a reasonable throughput and cost [35].

Genetic screening in *Caenorhabditis elegans* has effectively identified drug targets for human diseases ranging from depression (*e.g.*, Prozac) [36] to metabolic disease (*e.g.*, metformin) [37]. As the first, and only, model organism enabling whole-genome systemic RNA interference (RNAi) *in vivo* through feeding, *C. elegans* is an ideal system for high throughput identification of gene function [38,39]. *C. elegans* is evolutionarily distant from humans. Nevertheless, core lipid, sugar, and protein metabolism pathways are conserved between the two species [40]. Regulators such as TOR kinase and AMPK, and transcription factors such as Sterol response element binding protein (SREBP), Peroxisome proliferator-activated receptor gamma (PPARγ), and Transcription Factor EB (TFEB), similarly control metabolic genes and cellular responses to nutrients in both organisms. Loss of function of such regulators causes similar metabolic dysregulation, such as obesity or resistance to it, in worms and mammalian systems [41–45]. Moreover, in terms of identifying druggable targets, an obesity candidate gene identified in human GWAS whose ortholog is demonstrated to contribute to obesity in *C. elegans* is more likely to be a robust anti-obesity target across human populations.

As for diet-induced obesity (DIO), a high-glucose DIO model that shows both increased fat accumulation and shortening of lifespan has been previously described for *C. elegans* [46,47]. Less unanimous, though, are previous reports on the effect of fructose supplementation on the *C. elegans* diet. A diet supplemented with 50mg/mL or 100mg/mL of fructose was shown to reduce *C. elegans* healthspan and lifespan by Lodha *et al* [48] and Zheng *et al* [49], respectively. However, fructose at doses of 10 and 20mg/mL extended *C. elegans*'s lifespan. Further, the obesogenic effect of dietary supplementation of fructose has not been previously demonstrated in *C. elegans*.

In this study, we exploit a high-throughput *in vivo* obesity screen system to test for causality genes significantly associated with obesity in human GWAS. First, we identified 340 candidate genes from published GWAS and built a *C. elegans* RNAi library containing 293 worm orthologs of the human-obesity candidate genes. We used our previously developed screening pipeline [39,50] to perform an RNAi screen for genes whose inactivation alters the fat content of *C. elegans* fed a regular diet (RD). In this screen, we found 14 obesity genes (inactivation leads to obesity) and two lean genes (inactivation leads to leanness). We also established a *C. elegans* DIO model by feeding worms a high-fructose diet (HFrD). We show that worms fed a HFrD not only have higher fat content and body size, but also exhibit shortened health and lifespan. Using this DIO model, we identified three human genes whose *C. elegans* orthologs are DIO suppressors (inactivation prevents HFrD-induced obesity). Furthermore, we show that inactivation of the three DIO suppressors also ameliorates the detrimental effects of a HFrD on *C. elegans* health and lifespan. Altogether, this study provides a path to validate human GWAS obesity candidates *in vivo* in a high-throughput manner for future development of pharmacological interventions to reduce the burden of obesity.

## Results

### Meta-analysis of obesity GWAS variants identifies 207 human obesity candidate genes with orthologs in *C. elegans*

To identify genes that potentially contribute to human obesity, we exploited published meta-analyses of genome-wide association studies (GWAS) of obesity traits [51,52]. Together, three

studies report over 1,200 loci associated with an increased body-mass index (BMI), waist/hip ratio (a measure of fat distribution), or other metabolic traits. However, as is standard for these kinds of studies [53–55], the majority of the variants were not linked to genes. Therefore, we searched through the three previous studies for high-quality candidate genes linked to obesity loci. First, work by our group used eQTL analysis of 770 subcutaneous adipose samples from the Metabolic Syndrome in Men (METSIM) study [56] to link about 680 of these loci to 211 obesity gene candidates (S1 Table). We previously found genetic variants that were associated with both the expression of the candidate gene in subcutaneous adipose tissue and with various metabolic GWAS traits. For the present study, we considered a subset of 211 genes that are associated with obesity-related traits only. We performed Mendelian Randomization analyses to confirm that these genes are likely causal for obesity. Secondly, 120 novel gene candidates were taken based on the genomic location from a study that searched for novel loci associated with BMI in a trans-ancestral meta-analysis of 173,430 samples [57] Finally, Chu *et al*. [58] analyzed 2.6 million SNPs in up to 9,594 women and 8,738 men of European, African, Hispanic, and Asian ancestry, and based on the genomic location, they predicted 11 novel genes as associated with ectopic fat accumulation in the cohort [58], which we added to our candidate list. Together, we extracted 340 novel genes associated with obesity traits in humans from the three studies (S1 Table, [56–58]). Next, using the comparative genomic analysis tool Ortholist2 [59], we defined that 207 out of the 340 human candidate genes (67%) had orthologs in *C. elegans* (Fig 1A and S1 Table). However, in some cases, a human gene had more than one *C. elegans* ortholog; therefore, we identified a total of 386 *C. elegans* orthologs that corresponded to the 207 human gene candidates (Fig 1A and S1 Table). We then moved to use *in vivo* functional genomics screening to determine which of the human-obesity candidate genes have causal relationships with fat storage in *C. elegans*.

## 16 human-obesity gene candidates modulate fat storage in standard-diet fed *C. elegans*

Although we identified 386 *C. elegans* orthologs of 207 genes associated with obesity traits in humans, only 293 of these 386 ortholog genes were available in the Ahringer (original and supplementary) [60] or the Vidal [61] *C. elegans* genome-wide RNAi libraries (Fig 1A and S1 Table). Therefore, we built an RNAi sub-library consisting of these 293 worm genes for screening, which corresponded to 187 human-obesity gene candidates (workflow in S1A Fig). To increase the efficiency of the RNAi knockdown we used the RNAi-hypersensitive mutant *rrf-3* (*pk1426*) as the genetic background for the screen [62], after confirming it has wild type quantity and distribution of fat as made evident by the fat-specific dye Oil Red O (ORO) (S1B Fig). Although the loss of *rrf-3* function makes *C. elegans* hypersensitive to RNAi (including in neurons, [62]), it is important to keep in mind that some gene targets may not be accessible to RNAi even in this genetic background. Further, gene-gene and gene-environment interactions may be altered by mutation of *rrf-3*. As such, the phenotypic outcomes of our screen are conditioned by the choice of genetic background (*rrf-3*), which may alter the phenotype of genes with background-specific phenotypic effects (*e.g.*, only evident in metabolically sensitized backgrounds [63,64] or in the wild type background).

RNAi treatments, fat staining with ORO, and imaging for the screen were performed in duplicate and retested in three independent biological replicates as described in the Materials and Methods section. Using Cellprofiler feature extraction coupled to Cellprofiler Analyst classification of the microphotographs of fat-stained worms, RNAi treatments were categorized as follows: (1) wild type: >50% of the worms in the well showed quantities and distribution of ORO signal indistinguishable from that of animals treated with empty vector RNAi control

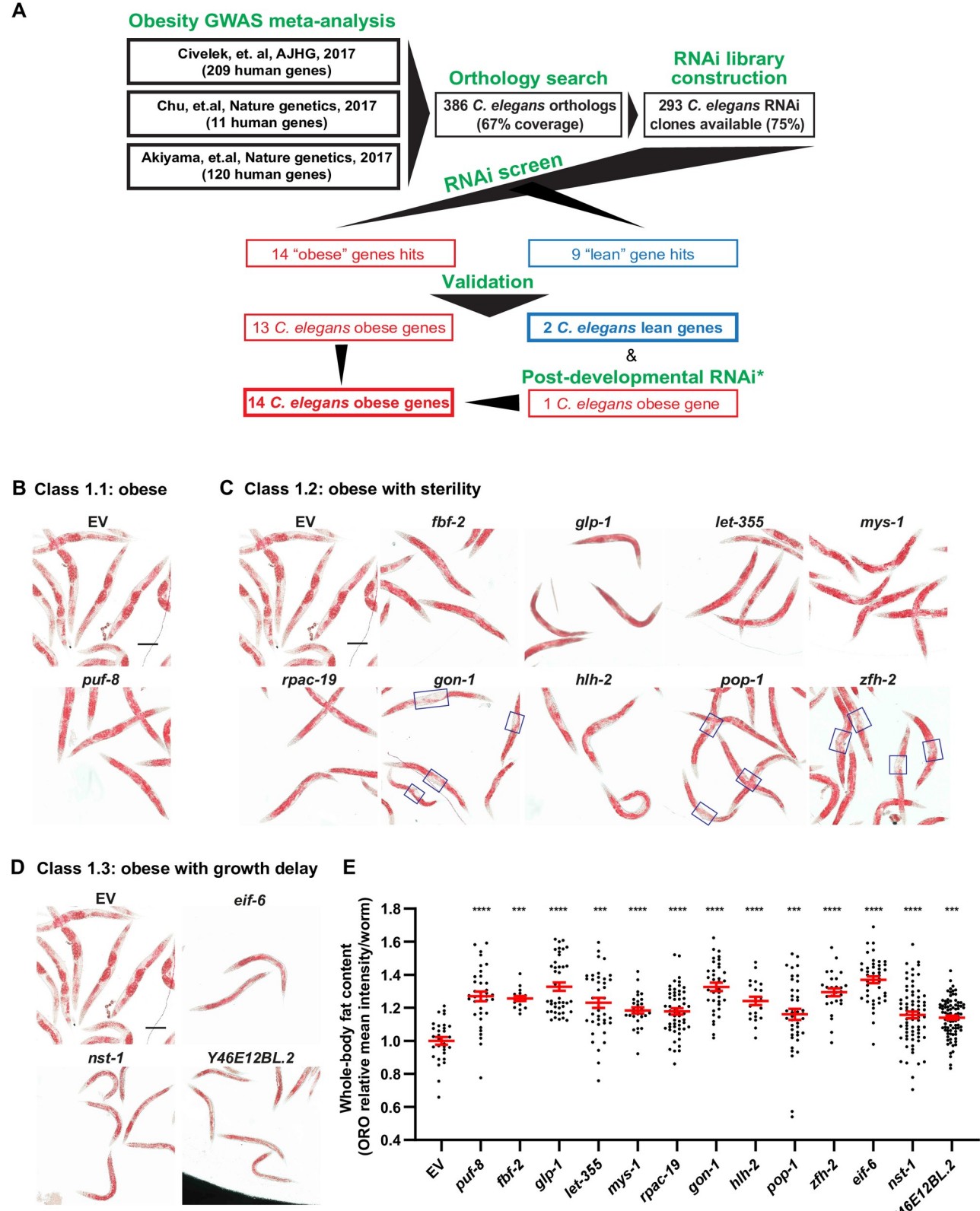

**Fig 1. Human obesity GWAS loci are causal obesity genes in *C. elegans*.** Throughout this figure: Error bars = S.E.M. N = numbers of independent biological replicates. *p≤ 0.05, **p≤0.01, ***p≤0.001 and ****p≤0.0001. (**A**) Summary of the meta-analysis and *C. elegans* RNAi screen results.

GWAS obesity candidate genes from three studies [56–58] were selected for RNAi screen. (**B-D**) Representative images of empty vector RNAi controls (EV) and RNAi-treated worms stained with oil red O (ORO). All images are in the same magnification, scale bar = 200μm. Hit genes were categorized into 3 groups: (**B**) Obese without obvious pleiotropies; (**C**) Obese with sterility. Blue boxes denote unusual "fat depletion around the vulva" phenotype; and (**D**) obese with developmental delay. (**E**) Quantification of ORO signal intensity in RNAi-treated worms for all treatments shown in panels B-D. Each data point represents the measurement of a single worm relative to the EV mean. N = 3. Ratio t-test was used to make comparisons between EV and the treatments.

(EV); (2) obese: after RNAi treatment >50% of the worms in these populations showed more intensity or broader distribution of the ORO signal than the parallel EV controls, suggesting that the gene is necessary to prevent obesity in animals fed a standard diet; and (3) lean: >50% of the worms in the RNAi-treated population showed lesser intensity or narrower distribution of the ORO signal than the parallel EV controls, suggesting that the gene is necessary to store normal levels of fat in *C. elegans* fed a standard diet. The gene knockdowns that exhibited a consistent lean or obese phenotype in three independent replicates were selected for further validation (Fig 1A and S1 Table).

We identified 23 genes altering fat storage in the primary screen carried out in 96-well plates. To validate the causal role of these genes in *C. elegans* reared in standard conditions, we retested the RNAi-driven phenotypes in worms grown in 6cm RNAi plates. In the retest, as in the primary screen, animals were treated with RNAi from the L1 stage. In this validation setup, ORO signal per worm was quantitated using ImageJ. We confirmed 13 out of 14 of the primary obese-gene hits: *puf-8* (human gene *PUM2*), *fbf-2* (*PUM2*), *glp-1* (*NOTCH4*, *EYS*), *let-355* (*DHX33*), *mys-1* (*KAT8*), *rpac-19* (*POLR1D*), *gon-1* (*ADAMTS9*), *hlh-2* (*TCF12*), *pop-1* (*TCF7L2*), *zfh-2* (*ZFHX3*), *eif-6* (*EIF6*), *nst-1* (*GLN3*), and *Y46E12BL.2* (*RRP12*) (Fig 1B, 1C, 1D and 1E and S1 Table). Of these obese genes, knockdown of all but *puf-8* also caused sterility; hence, hereinafter we refer to these hits as sterile genes (Fig 1C). However, the seemingly exceptional phenotype of *puf-8* may be due to our binary sterile versus fertile classification. In fact, *puf-8* deficiency has been linked to germline phenotypes including reduced germline, reduced progeny size, and a tumorous germline [65,66]. If confirmed with a null mutant, the quantitative (as opposed to qualitative) fertility phenotype of *puf-8* opens the exciting possibility of using it to more finely dissect the mechanisms determining soma versus germline nutrient allocation. Furthermore, RNAi against *gon-1*, *hlh-2*, *pop-2*, and *zfh-2* promoted an unusual sterility phenotype, in which animals showed extensive fat depletion around the vulva (Fig 1C), suggesting a possible change in fat tissue distribution when germline/eggs are lacking that also warrants future study. Distinctively, *eif-6*, *nst-1*, and *Y46E12BL.2* showed a developmental-delay phenotype (Fig 1D). A link between obesity and infertility, and obesity and compromised growth, has been extensively documented [40,67–69]. It would then be important for future studies to define whether obesity is the cause or consequence of these phenotypes.

The lean hits in the primary screen were *rpt-5* (human gene *PSMC3*), *hsp-4* (*HSPA1B*), *let-767* (*HSD17B12*), and *Y71H10B.1* (*NT5C2*). Similar to the obese genes, the lean genes could also be subclassified. Animals treated with RNAi against *Y71H10B.1* showed body fat reduction without developmental delay (Fig 2A and 2B). By contrast, RNAi-driven inactivation of *rpt-5*, *hsp-4*, and *let-767* led to severe developmental delay (S1C Fig, and S1 Table). As previously reported [70], we observed that body fat content increased with age (S1D Fig). Therefore, we sought to test whether we could uncouple the developmental delay from the fat phenotype by starting the RNAi treatment at the L4 instead of the L1 stage. In animals fed RNAi from the L4 stage, we found: (1) No fat phenotype for *hsp-4* (S1E and S1F Fig), suggesting increased resilience to ER-stress in the adult *C. elegans*; (2) Reduced fat content after inactivation of *let-767* (Fig 2A and 2C), showing that the function of this gene in fat metabolism can be

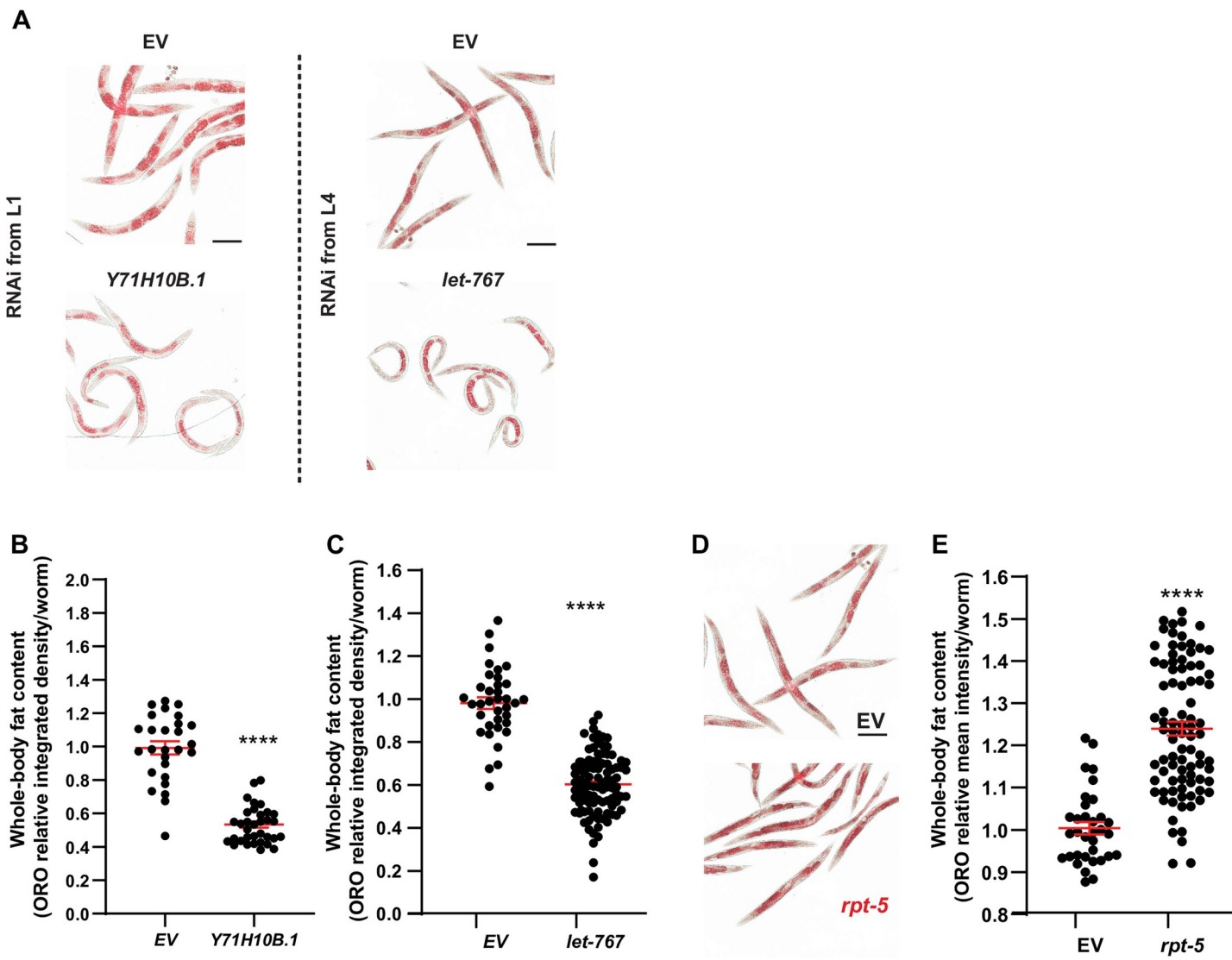

**Fig 2. Knock down of *Y71H10B.1/NT5C2* and *let-767/HSD17B12* promotes leanness in *C. elegans*.** Throughout this figure: Scale bars = 200μm. Error bars = S.E.M. N = number of independent biological replicates. Statistical significance was assessed via ratio t-test, *p≤ 0.05, **p≤0.01, ***p≤0.001 and ****p≤0.0001. (**A**) Representative images of the body fat content and distribution in worms treated with RNAi against *Y71H10B.1/NT5C2* (from the L1 stage) and *let-767/HSD17B12* (from the L4 stage because *let-767* RNAi from the L1 stage results in severe developmental arrest; S1C Fig). (**B-C**) ORO signal intensity in RNAi-treated worms for the treatments shown in panel **A.** Each data point represents the measurement of a single worm relative to the EV mean. N = 3. (**D**) Representative image of the body fat content and distribution in worms treated with RNAi against *rpt-5/PSMC3* (from the L1 stage). (**E**) ORO signal intensity in *rpt-5* RNAi-treated worms. Each data point represents the measurement of a single worm relative to the EV mean. N = 3.

uncoupled from its role in development, and suggesting that this gene may impair development via metabolic dysfunction; and (3) Surprisingly, although knockdown of *rpt-5* from the L1 stage impaired fat accumulation, knockdown of this same gene from the L4 stage caused obesity (compare S1C Fig to Fig 2D and 2E), providing an extreme example of context-dependent gene function. Therefore, altogether, we identified 14 obese genes–*puf-8*(*PUM2*), *eif-6* (*EIF6*), *fbf-2*(*PUM2*), *glp-1*(*NOTCH4, EYS*), *gon-1*(*ADAMTS9*), *hlh-2*(*TCF12*), *let-355* (*DHX33*), *mys-1*(*KAT8*), *nst-1*(*GLN3*), *pop-1*(*TCF7L2*), *rpac-19*(*POLR1D*), *rpt-5*(*PSMC3*), *Y46E12BL.2*(*RRP12*), *and zfh-2*(*ZFHX3*)–and 2 lean genes–*Y71H10B.1*(*NT5C2*) and *let-767* (*HSD17B12*) (Summary in Fig 1A and S1 Table).

The 16 confirmed worm hits correspond to 16 unique human genes (S1 Table). The phenotypes in *C. elegans* suggest these 16 genes may have a conserved causal role in obesity. In the absence of monogenic human diseases or knock-out mice lines, we decided to approximate a test of this hypothesis using three publicly available datasets: (1) Gene expression data from adipose tissue from the Hybrid Mouse Diversity Panel [71] (HMDP)–a cohort of well-phenotyped male and female mice fed a high-fat diet; (2) Subcutaneous adipose tissue gene expression data from the METSIM cohort–a thoroughly phenotyped cohort of Finnish men [56]; and (3) the GWAS catalog (www.ebi.ac.uk/gwas/home). With the HMDP and METSIM datasets, we could define whether reduced or increased expression of the human obesity candidate genes (or their mouse orthologs) were associated with obesity. The GWAS catalog enabled us to investigate to what extent the gene may be specifically affecting metabolic disease, as opposed to promoting more general sickness with an indirect impact on body-mass index (BMI) or waist-hip ratio (WHR).

In the HMDP cohort, we found 12 mouse orthologs of the *C. elegans* hits (*Adamts9*, *Dhx33*, and *Kat8* were not measured in this study). Remarkably, 7 out of the 8 genes whose expression negatively correlates with obesity traits in the mouse, also lead to obesity when they are knocked down via RNAi in *C. elegans* (Fig 3A–bolded names in the gene set to the left), suggesting that the mouse variants would lead to loss or reduced function of these genes and that these seven genes would play similar roles in the regulation of fat storage in *C. elegans* and the mouse. Specifically, lower expression of *Pum2*, *Polr1d*, *Notch4*, *Tcf7l2*, *Tcf12*, *Psmc3*, and *Zfhx3* correlates with increased body fat percentage, fat mass, body weight, and total mass in the mouse, which is comparable to knockdown of the orthologs *puf-8*, *fbf-2*, *rpac-19*, *glp-1*, *pop-1*, *hlh-2*, *rpt-5*, and *zfh-2* causing obesity in *C. elegans*. On the other hand, we found *Nt5c2* expression to positively correlate with obesity traits in the mouse, which is in line with the knockdown of the *C. elegans* ortholog–*Y71H10B.1*– leading to leanness in *C. elegans* (Fig 3A–bolded name in the gene set to the right).

To test the hypothesis that the *C. elegans* hit genes would have conserved causal roles in human obesity we performed similar analyses using the METSIM cohort. We found 16 human genes corresponding to 16 *C. elegans* gene hits in this dataset. However, on one hand, *NOTCH4* and *EYS* are both orthologs of *C. elegans glp-1*, and on the other hand, *puf-8* and *fbf-2* are both orthologs of human PUM2. We calculated the degree of association between the expression in adipose tissue of the 16 human genes and clinical traits including body mass index (BMI) and waist-to-hip ratio (WHR). BMI was the most strongly associated trait (Fig 3B). Further, most human genes showed expression associations with BMI that were in the same direction as in *C. elegans*. Specifically, lower expression of *PUM2*, *POLR1D*, *ADAMTS9*, *DHX33*, *NOTCH4*, *EYS*, *TCF7L2*, and *KAT8* correlates with increased BMI, which is comparable to the obesity phenotype observed in *C. elegans* after RNAi knockdown of their orthologs *puf-8*, *fbf-2*, *rpac-19*, *gon-1*, *let-355*, *glp-1*, *pop-1*, and *mys-1*. On the other hand, higher *NT5C2* expression positively correlates with obesity (Fig 3B), which is in line with the lean phenotype observed in *C. elegans* after RNAi-driven knockdown of the ortholog gene *Y71H10B.1*. A parsimonious hypothesis derived from these correlations would be that if RNAi against a *C. elegans* gene leads to obesity, and a low-expression variant of its ortholog gene is associated with increased BMI in humans, then the ortholog of the *C. elegans* hit gene is likely to have a causal role in human obesity. Although we appreciate the caveats of the premises leading to this hypothesis, including that changes in expression may be compensatory and not causal mechanisms, we present these analyses as an approximation to the likelihood that the human and *C. elegans* genes have causal roles in obesity in both organisms. Therefore, as for the most part, the expression associations with BMI in humans go in the same direction as they do in mice

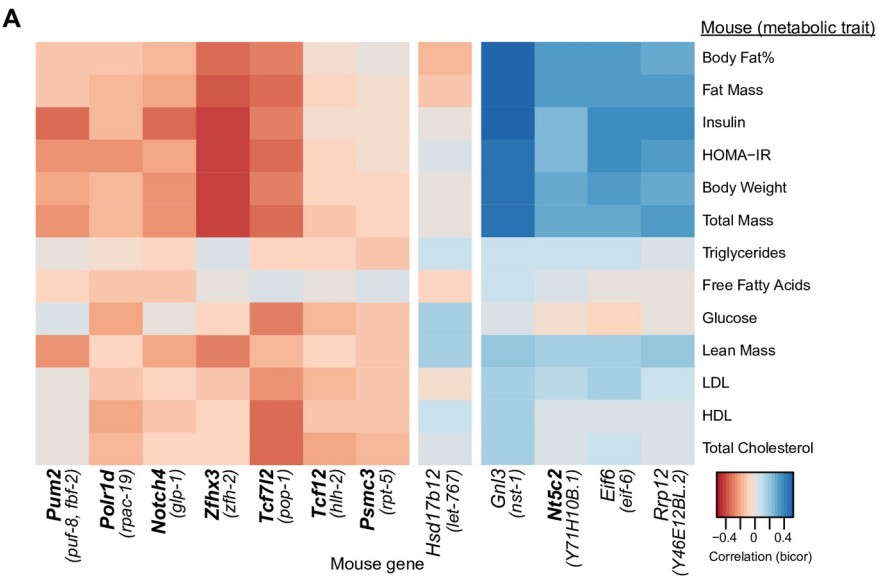

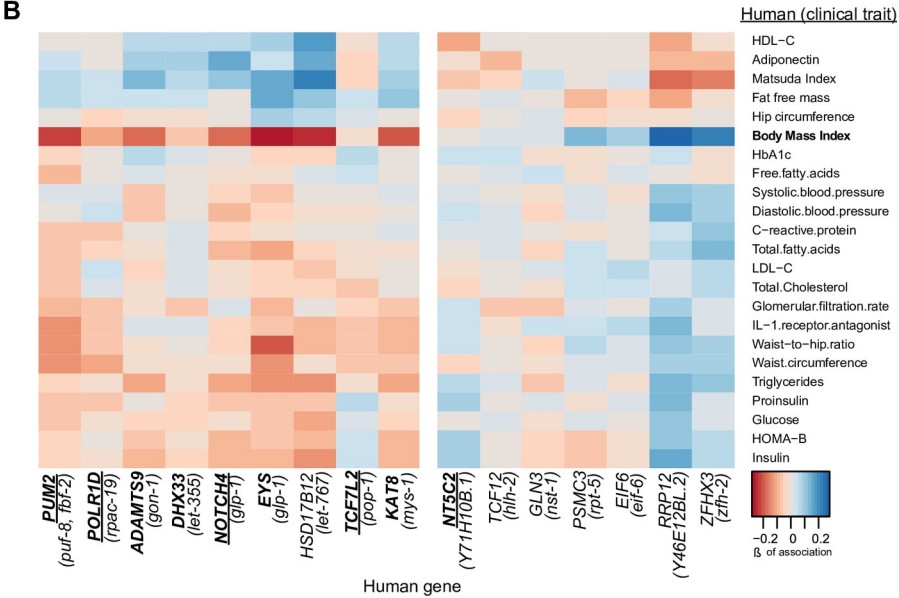

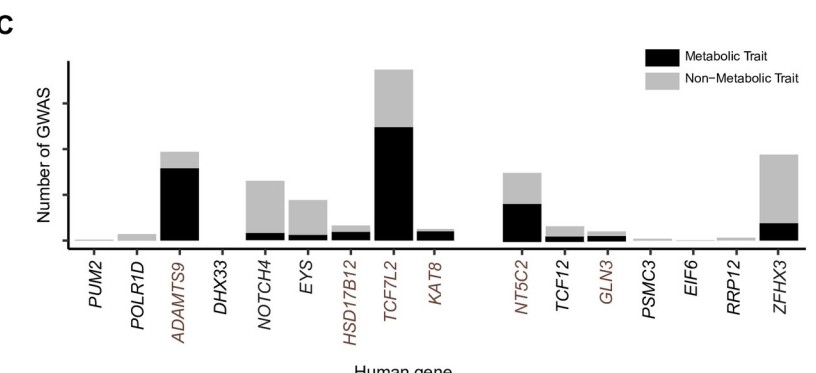

**Fig 3. Gene expression–disease phenotype associations are conserved for most *C. elegans* screen hits.** (**A**) Bi-weight Mid-correlations (median based correlation metric) of mouse gene expression and metabolic traits in adipose tissue of the HMDP cohort. Genes negatively associated with fat mass are shown on the left, and positive associations on the right. Bolded mouse ortholog names indicate that the expression of the mouse and the worm genes are associated with fat storage (fat mass in mice) in the same direction. (**B**) Association (Beta of association) between human gene expression in the subcutaneous adipose tissue and clinical traits in the METSIM cohort. Genes negatively associated with BMI are shown on the left. Positive associations are shown on the right. Bolded human ortholog gene names indicate that the expression of the human and the worm gene are associated with fat storage (BMI in humans) in the same direction. Underlined human ortholog names indicate the same correlation sign is observed in worms, mice, and humans. (**C**) Number of published metabolic and non-metabolic trait GWAS associations for each human ortholog present in the GWAS catalog. The assumption to interpret these data is that genes with variants mostly associated to metabolic traits would more likely be directly involved in metabolic or metabolic-related functions, whereas genes with GWAS variants in mostly non-metabolic traits are more likely to have pleiotropic effects including indirect weight loss or gain. Colors distinguish association with metabolic (black) and non-metabolic (grey) traits.

and *C. elegans*, we hypothesize that the orthologs of the *C. elegans* hit genes are likely to have conserved causal roles in human obesity.

Lastly, a gene variant can impact fat accumulation rather directly, because the gene is involved in fat metabolism or its regulation, or rather indirectly because the variant causes general sickness. We investigated this by defining to what extent a gene variant affects metabolic-specific or metabolic-related traits as opposed to unrelated disease traits. We queried the GWAS catalog for each human ortholog of the *C. elegans* hit genes. It is worth noting that the METSIM study has not yet been included in the GWAS catalog, therefore it was excluded from this analysis. Variants near *KAT8*, *TCF7L2*, *DHX33*, *POLR1D*, *HSD17B12*, *NT5C2*, and *MTCH2* were significantly associated with BMI. Variants near *GNL3*, *NOTCH4*, *ADAMTS9*, and *EIF6* were significantly associated with waist/hip ratio adjusted for BMI. (S1 and S2 Locus-Plots). We found 15 out of the 16 loci associated with any human disease traits (*DHX33* is unassociated). Of these, 11 genes were associated with metabolic traits beyond obesity such as insulin resistance, fat distribution, and cardiovascular disease in human GWAS (Fig 3C, black bars). Further, the variants linked to *ADAMTS9*, *HSD17B12*, *TCF7L2*, *KAT8*, *NT5C2*, and *GLN3* seem to be mostly associated with metabolic phenotypes as opposed to non-metabolic disease traits (Fig 3C, brown labels indicate genes with the largest proportion of metabolic traits), suggesting these genes may have a rather specific role in metabolic health. In summary, we found supportive evidence in mammalian systems of a role in obesity and metabolic syndrome for most of the orthologs of the genes that we causally linked to obesity in *C. elegans*.

## High-fructose diet leads to diet-induced obesity in *C. elegans*

Excessive dietary intake of fructose has been suggested to be a major driver of the obesity epidemic [72,73], as fructose is the most common additive in industrialized foods [74]. To test the potential contribution of the human GWAS obesity candidates in the development of DIO we established a fructose-driven *C. elegans* model of DIO. We named high-fructose diet (HFrD) the dietary regimen in which worms are grown from the L1 stage in plates of Nematode growth media (NGM) supplemented with 10mg/mL of fructose (1% fructose). We observed that at the adult stage, worms fed a HFrD show a significant increase in body fat content compared to those fed a regular diet (RD) (Fig 4A and 4B). Further, worms fed the HFrD are also larger than worms fed RD (Fig 4C), which is similar to a previous report showing increased body size in worms fed excessive glucose [75].

Adipose is the main fat storage tissue in humans, and adipocytes with increased lipid droplet (LD) number and size are cellular hallmarks of obesity [76]. In *C. elegans*, there are no specialized adipose cells. The primary triglyceride depots are found in the worm's intestinal cells and are contained in LDs sized between 0.5–1.5 μm [77,78]. To define whether *C. elegans* fed a

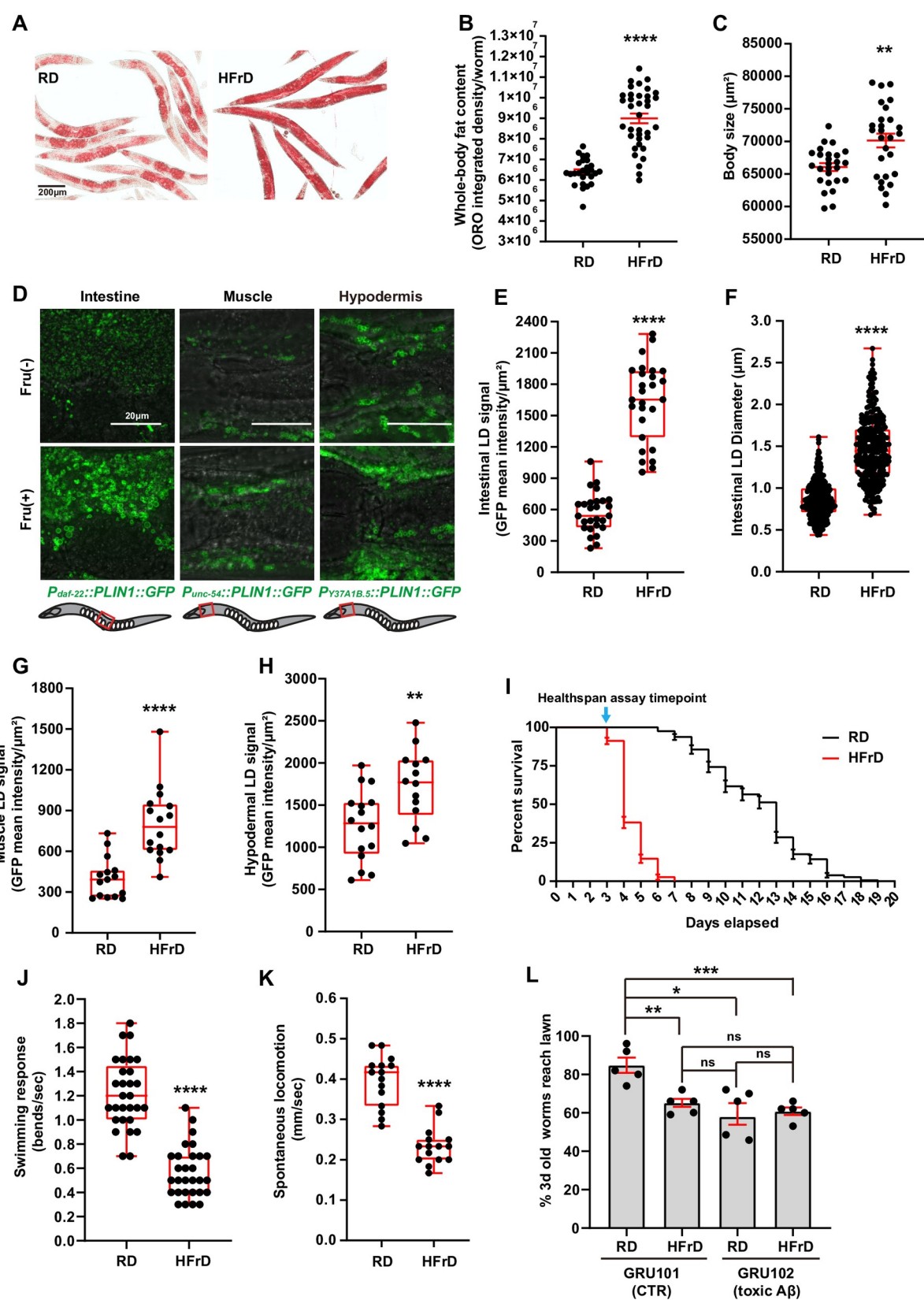

**Fig 4. A high-fructose diet leads to diet-induced obesity in *C. elegans*.** Throughout this figure: Error bars = S.E.M. N = numbers of independent biological replicates. Unless specified, unpaired nonparametric t-tests were used to assess the significance. $^*p \leq 0.05$, $^{**}p \leq 0.01$, $^{***}p \leq 0.001$ and $^{****}p \leq 0.0001$. (**A**) Representative images of the body fat content and distribution in worms fed regular diet (RD) or high-fructose diet (HFrD) from the L1 stage, and stained worms with ORO when day-1 adults. Scale bar = 200μm. (**B**) Quantification of ORO intensity in worms for fed RD or HFrD as represented in panel A. Each data point represents the measurement of a single worm. N = 3. (**C**) Body size quantification of worms fed RD or HFrD. Each data point represents the ImageJ-measured length of a single worm. N = 3. (**D**) Tissue-specific LD phenotype in worms fed RD (top) or HFrD (bottom). Description of the transgenic reporters is depicted in the panel. Scale bar = 20μm. (**E**) Mean GFP intensity of intestinal LDs per worm, in worms fed RD or HFrD as represented in D. Each data point represents the measurement of a 0.004mm$^2$ intestinal square area of a worm. N = 3. (**F**) Mean intestinal LD size in worms fed RD or HFrD. Each data point represents the size measurement of an individual LD randomly selected from ≥10 worms from each independent biological replicate. N = 3. (**G**) Mean GFP intensity of muscle LDs in worms fed RD or HFrD, as shown in D. Each data point represents the measurement of a 0.003mm$^2$ square area of muscle behind the pharynx of a worm. N = 3. (**H**) Mean GFP intensity of hypodermal LDs in worms fed RD or HFrD, as shown in D. Each data point represents the measurement of a 0.003mm$^2$ square area of hypodermis behind the pharynx of a worm. N = 3. (**I**) Lifespan assessment of worms fed RD or HFrD. N = 1 is shown in this panel, N = 3 summarized in S2 Table. (**J**) Locomotory capacity measured as body bends per second in worms fed RD or HFrD. N = 3. (**K**) Locomotory capacity measured as displacement velocity in worms fed RD or HFrD while on the surface of bacteria-free NGM plates. N = 3. (**L**) Neurolocomotory and sensorial capacity measured in a food race assay as described in Materials and Methods section. Genotypes: GRU101 = worms with pan-neuronal expression of non-toxic human Aβ, and GRU102 = worms with pan-neuronal expression of toxic human Aβ. N = 3.

HFrD would show changes in the abundance and/or size of the LDs, we used a LD reporter consisting of Drosophila PLIN1::GFP driven by the intestine-specific promoter *daf-22* [79,80]. In agreement with the mammalian subcellular hallmarks of obesity, we observed an increase in both the overall intensity (Fig 4D and 4E) and the size (Fig 4F) of the LDs in the intestine of HFrD-fed worms. Another common feature of human obesity is the increase in ectopic fat stores (fat outside the primary fat storage tissues) [81], with excess ectopic fat being associated with worse health outcomes [82]. Although *C. elegans* does not have a dedicated adipose tissue, fat beyond the intestine is observed in mutants with metabolic dysregulation such as the mTORC2-component mutant *rict-1*, in which fat is additionally observed in muscle and hypodermis [83,84]. To further characterize the distribution of the fat stores in the HFrD worms, we used the Drosophila PLIN1::GFP construct driven by the *unc-54* promoter to express it in the muscle, or driven by the *Y37A1B.5* promoter to express it in the hypodermis [79,80]. We again observed a significant increase in the LD intensity (Fig 4D, 4G and 4H) but no change in LD size (S3A and S3B Fig), in the muscle and hypodermis of worms fed HFrD. These observations suggest excess fat is stored in primary and ectopic tissues in *C. elegans*, comparable at a subcellular level to what is observed in visceral and ectopic fat depots in humans with obesity.

Obesity is defined by the World Health Organization as abnormal or excessive fat accumulation that presents a higher risk of debilitating co-morbidities and death [85–87]. In *C. elegans*, excessive fat accumulation is not always associated with deleterious co-morbidities in health. For instance, mutation of the *C. elegans* insulin receptor (IR) *daf-2* leads to excessive fat accumulation [67]. However, *daf-2* mutant animals are resistant to age-related decline and long lived [88–90]. Consistently, even though GH receptor knockout (GHRKO) mice have increased fat mass and decreased lean mass, they show improved insulin sensitivity and are long lived [91]. Therefore, to establish an informative model of DIO in *C. elegans*, it is critical to test whether the increase in fat levels correlates with detrimental effects on health. To this end, we first evaluated the effect of HFrD on overall survival by comparing the lifespan of HFrD-fed worms to the lifespan of the RD-fed worms. HFrD reduced the median lifespan of *C. elegans* by 69% (Gehan-Breslow-Wilcoxon test), with a 31.49 average Hazard Ratio (Mantel-Haenszel test) (Fig 4I and S2 Table). Next, we conducted healthspan assays as described previously [90]. In the case of locomotory capacity, we found a significant reduction in body bending rate and average velocity in 3-day old worms fed HFrD when compared to worms of the same age fed RD (Fig 4J and 4K). In *C. elegans*, locomotion defects including impaired body bending and reduced velocity can be caused by reduced proteostasis [92], and in humans,

obesity is strongly associated with neurodegenerative diseases characterized by uncurbed protein aggregation [93–95]. Therefore, we hypothesized that high fructose levels in the *C. elegans* diet may reduce proteostasis, which could in turn lead to an earlier onset of locomotion defects. As previously reported, *C. elegans* constitutively expressing a toxic form of the human Aβ amyloid (strain GRU102) show an earlier onset of locomotory impairment than worms expressing a non-toxic form of the human Aβ amyloid (strain GRU101), and this reduced locomotory capacity is due to reduced proteostasis in the neuro-locomotory system [92,96]. To test the hypothesis that reduced proteostasis would contribute to the reduced locomotory capacity observed in worms fed HFrD, we fed a regular diet or HFrD to the GRU101 and GRU102 worms. We conducted a "food race" assay as previously described [97]. Briefly, we placed fifty 3-day old adult worms from each condition (GRU102 ± fructose and GRU101 ± fructose) at one end of a 10cm NGM plate. To start the assay, 25μl of *E. coli* suspension ($OD_{600nm}$ = 20) were placed on the plates on the opposite side of the worms. After 1h, the number of the worms fully or partially within the borders of the mini bacterial lawn was counted. We consistently found fewer worms expressing toxic Aβ than worms expressing the non-toxic Aβ in the lawns (Fig 4L), confirming the published detrimental effect of increased protein aggregation on *C. elegans* locomotion. We also observed HFrD to be sufficient to impair *C. elegans* locomotion (Fig 4L). Tellingly, the HFrD-derived impairment was not additive to the expression of toxic Aβ (Fig 4L), suggesting the possibility that HFrD and Aβ overexpression may share a common mechanism of toxicity, which we hypothesize might be reduced proteostasis. Altogether, we found that overconsumption of fructose in *C. elegans* evokes several of the hallmarks of human obesity including elevated body fat content in primary and ectopic fat storage tissue subcellularly characterized by increases in the number and size of the LDs, in conjunction with reduced health and lifespan.

## Three human-obesity gene candidates cause DIO in *C. elegans*

We tested the 293 *C. elegans* orthologs of the human obesity GWAS candidate genes for DIO causality by screening for RNAi treatments that led to reduced or no obesity even when worms were fed excessive fructose. The screening set up for this DIO screen is as described for the regular-diet screen (S1A Fig) except that 10mg/mL of fructose (1% fructose) were supplemented to the Nematode growth media plates. After knocking down the 293 *C. elegans* genes, we observed two phenotypic classes: (1) Wild type: after RNAi treatment, worms in these populations showed the larger quantity and broader distribution of ORO signal that characterizes DIO; this was the expected ORO phenotype because the worms were feeding excessive levels of fructose; and (2) DIO suppressors: ≥50% of worms in an RNAi treatment showed reduced fat content when compared to EV controls, suggesting that the gene normally contributes to HFrD-driven obesity in *C. elegans*. The RNAi treatments that showed a DIO-suppressor phenotype consistently in 3 or more independent biological replicates were selected for further validation.

From the primary DIO screen, we identified eight high-confidence DIO-suppressor genes. Using the same rationale and approach described above (S1A Fig), we retested the eight genes in 6cm NGM plates. We confirmed the DIO-suppressor phenotype for five out of the eight primary hits. From the five DIO suppressors, *pho-1* (human gene *ACP2*) and *Y71H10B.1* had no detrimental effects on development (Fig 5A, 5B and 5C). However, the other three genes–*rpt-5*, *hsp-4*, and *let-767* –caused severe developmental delay (S3C Fig and S1 Table). Starting the RNAi treatments against these genes at the L4 stage, showed that post-developmental inactivation of *rpt-5* and *hsp-4* did not prevent DIO (S3D Fig). On the other hand, L4-knockdown of *let-767* reduced body fat content and body size in worms fed HFrD (Fig 5A, 5D and 5E),

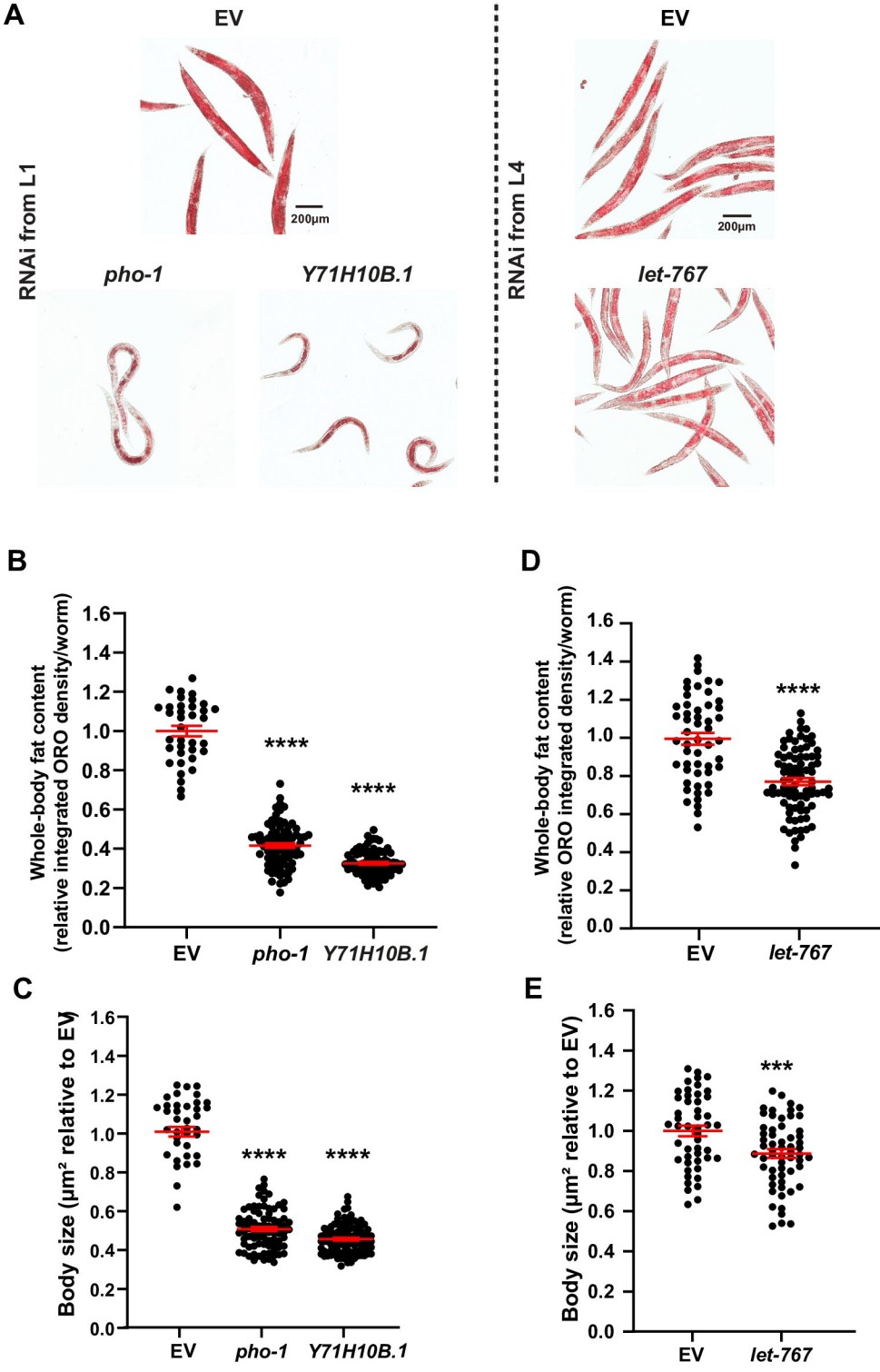

**Fig 5. *Y71H10B.1/NT5C2*, *let-767/HSD17B12*, and *pho-1/ACP2* inactivations prevent diet-induced obesity in *C. elegans*.** Throughout this figure: Scale bars = 200μm. Error bars = S.E.M. N = number of independent biological replicates. Statistical significance was assessed via ratio t-test, *p≤ 0.05, **p≤0.01, ***p≤0.001 and ****p≤0.0001. (**A**) Representative images of the body fat content and distribution observed in worms fed HFrD while treated with RNAi against the following genes: Left = *pho-1* and *Y71H10B.1* RNAi treatment from the L1 stage. Right = *let-767* RNAi treatment from the L4 stage. (**B** & **D**) ORO signal intensity in worms treated with RNAi as shown in panel A. Each data point represents the ORO intensity values in each worm normalized to the mean ORO intensity value of the EV

control. N = 3. (**C & E**) Body size of worms treated with RNAi as shown in panel A. Each data point represents the measurement of a worm. Individual body size values were normalized to the mean value of the EV control. N = 3.

suggesting that *let-767* would independently modulate fat metabolism and development. Importantly, *let-767* and *Y71H10B.1* were also lean hits in the regular diet screen described above (Fig 2), suggesting a generic function for these genes in promoting fat accumulation. By contrast, *pho-1* knockdown only prevented obesity in worms fed a HFrD, suggesting a specific function in DIO. Together, we identified and confirmed three genes–*pho-1*, *let-767*, and *Y71H10B.1* –whose knockdown prevented the development of obesity in animals fed excessive sugar.

We already described above that expression of the human ortholog of *Y71H10B.1*, *NT5C2*, positively correlates with waist-to-hip ratio and circulating triglycerides (Fig 3B), which is in line with knockdown of *Y71H10B.1* in *C. elegans* promoting leanness in animals fed a regular or high-fructose diet. In a more difficult to interpret case, the expression of the human ortholog of *let-767*, *HSD17B12*, although negatively associated with BMI, is positively associated with free fatty acids, indicating that reduced expression of *HSD17B12* is associated with obesity but also with improved lipid metabolism. Suggesting a similarly complicated role, expression of the mouse ortholog, *Hsd17b12*, is both negatively (*e.g.*, with % body fat) and positively (*e.g.*, glucose and circulating triglycerides) associated with markers of metabolic health (Fig 3B). Therefore, future studies would be necessary to define whether the role of *let-767* in promoting obesity is conserved in mammals. Finally, using the METSIM cohort, we found that the expression of the human ortholog of *pho-1*, *ACP2*, positively correlates with body weight and total triglycerides (S4 Fig), which is in line with our observations showing that knockdown of *C. elegans*'s *pho-1* prevents diet-induced obesity.

## DIO suppressors restore *C. elegans* LD physiology in a tissue-specific manner

As described above, HFrD promotes increased accumulation of larger LDs in both primary and ectopic fat depots (Fig 4D). Using the PLIN1::GFP LD-reporter strains, we assessed whether knockdown of the generic lean genes *let-767* and *Y71H10B.1*, and the specific DIO-suppressor gene *pho-1*, would suppress the LD number and/or size phenotypes, in primary and/or ectopic tissues.

*C. elegans* treated with RNAi against *Y71H10B.1* showed smaller LDs in the intestine (Fig 6A and 6A'), and less overall LD signal in the intestine, muscle, and hypodermis when animals were fed a regular diet (Fig 6A, 6B, 6C and 6D). Similarly, RNAi against *Y71H10B.1* in HfrD fed worms reduced LD size in the intestine (Fig 6A and 6A'), and overall LD signal in the intestine, muscle, and hypodermis (Fig 6A, 6B, 6C and 6D). The results were, hence, in line with a diet-independent role for *Y71H10B.1* in the control of fat storage in *C. elegans*. On the other hand, although *C. elegans* treated with RNAi against *let-767* showed reduced LD size (Fig 6A and 6A'), the treatment did not significantly change LD intensity in intestine or muscle when *C. elegans* were fed RD (Fig 6A, 6B and 6D). By contrast, RNAi against *let-767* appreciably decreased LD-signal intensity in the hypodermis of animals fed RD (Fig 6C and 6D), suggesting that most of the overall loss of fat observed in *let-767* RNAi-treated animals fed regular diet is due to loss of hypodermal fat (Fig 6C and 6D). Distinctively, and showing the complexity of the fat-storage response to diet, *let-767* RNAi-treated worms fed HFrD showed reduced LD size and overall LD signal in the intestine, as well as decreased overall LD signal in the hypodermis (Fig 6A, 6C and 6E). By contrast, LD signal did not change in the muscle when

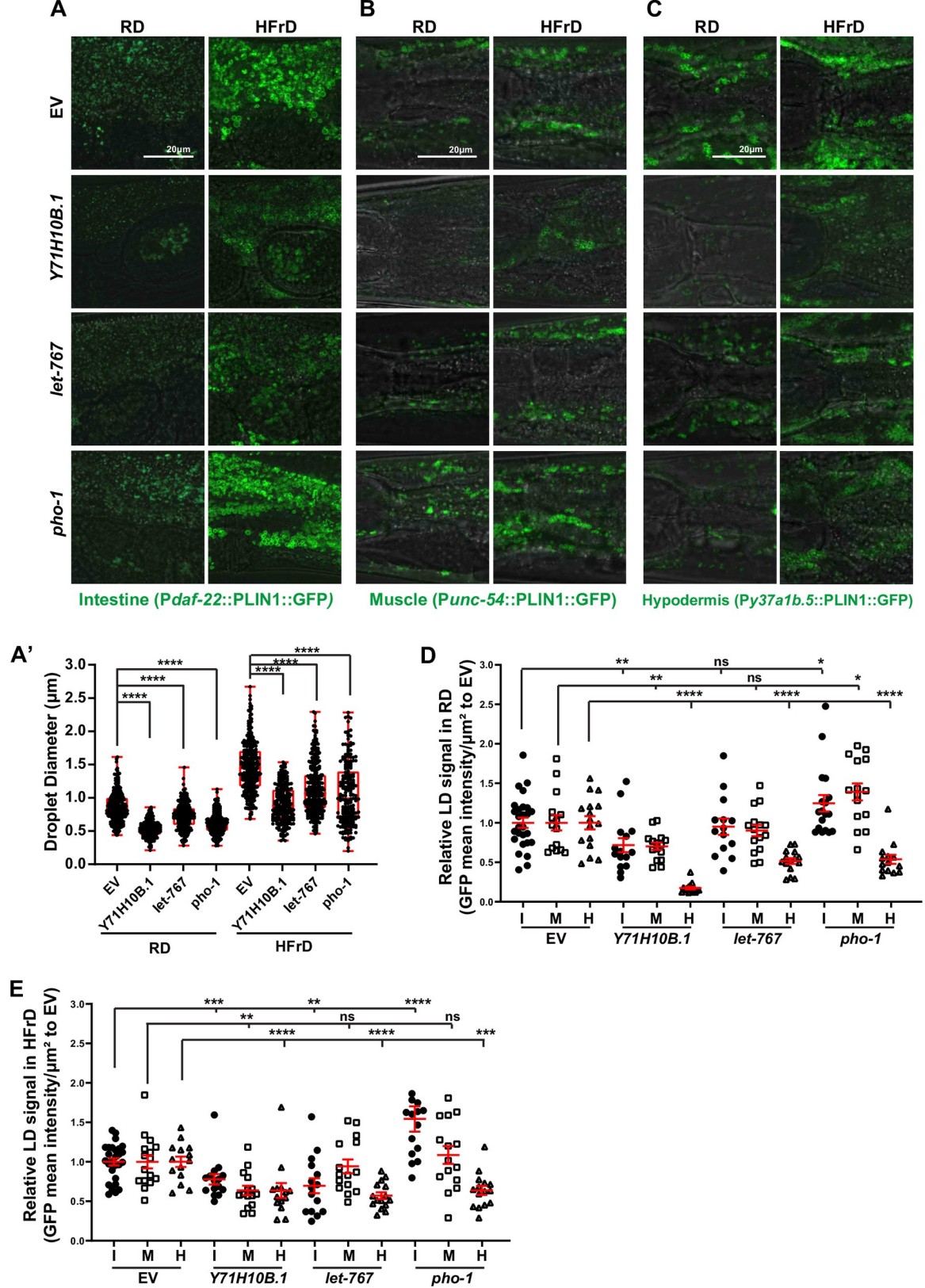

**Fig 6. Diet-induced obesity suppressors tissue-specifically restore lipid droplet homeostasis.** Throughout this figure: Scale bars = 20μm. Error bars = S.E.M. N = numbers of independent biological replicates. Unless specified, ratio t-tests were used to assess the

significance. *p≤ 0.05, **p≤0.01, ***p≤0.001 and ****p≤0.0001. Tissue-specific lipid droplet (LD) reporters strains XD3971, XD1875 and XD2458 express *Drosophila PLIN1::GFP* in the intestine (*daf-22* promoter), muscle (*unc-54* promoter), and hypodermis (*Y37A1B.5* promoter), respectively. LD size and abundance assessments as described in the Materials and Methods section. Each data point represents the GFP measurement of a 0.003mm$^2$ square area of the tissue. I: intestine, M: muscle, H: hypodermis. N = 3.Unpaired nonparametric t-tests were used to assess the significance. (**A-A'**) LD size in the *C. elegans* intestine is reduced in *Y71H10B.1*, *let-767*, and *pho-1* RNAi-treated animals fed regular or HFrD diet. (**A & D**) LD abundance in the intestine (I) of worms fed regular diet (RD) is reduced in animals treated with RNAi against *Y71H10B.1*, but not *let-767*. *pho-1* increases the LD pool in the intestine. N = 3. (**B & D**) Similar to the intestine, LD abundance in the muscles (M) of *C. elegans* fed regular diet (RD) is reduced in *Y71H10B.1*, but not in *let-767* RNAi-treated animals. *pho-1* RNAi increases the LD pool in the intestine. N = 3. (**C & D**) LD abundance in the hypodermis (H) of *C. elegans* fed regular diet (RD) is reduced in animals treated with RNAi against *Y71H10B.1*, *let-767*, and *pho-1*. N = 3. (**A & E**) LD abundance in the intestine of worms fed a high-fructose diet (HFrD) is reduced by RNAi treatment against *Y71H10B.1* and *let-767*, whereas RNAi against *pho-1* increases the LD pool in the intestine. N = 3. (**B & E**) LD abundance in the muscles of *C. elegans* fed HFrD is reduced in animals treated with RNAi against *Y71H10B.1*, but not *let-767* or *pho-1*. N = 3. (**C & E**) LD abundance in the hypodermis of *C. elegans* fed HFrD is reduced in animals treated with RNAi against *Y71H10B.1*, *let-767*, and *pho-1*. N = 3.

compared to WT worms fed HFrD (Fig 6B and 6E). Therefore, *let-767* contributes to fat accumulation in the hypodermis independent of the diet and in the intestine only in HFrD, whereas it does not contribute to fat accumulation in the muscle of *C. elegans*.

Screening and validation showed that *pho-1* contributes to DIO but not to basal fat accumulation in *C. elegans* (Figs 5A, S3C and S3D, and S1 Table). Therefore, it was initially surprising to find that regular diet-fed animals treated with RNAi against *pho-1* showed smaller LDs in the intestine (Fig 6A and 6A') and reduced overall LD signal in the hypodermis (Fig 6C and 6D). However, simultaneously, *pho-1*-treated worms showed a subtle increase in overall LD signal in the intestine and muscle (Fig 6A, 6B and 6D). Hence, we hypothesize that smaller LDs in the intestine and overall less LD content in the hypodermis, in conjunction with increased overall LD signal in intestine and muscle, may lead to a net fat signal in *pho-1* knockdown animals fed a regular diet that is indistinguishable from WT when scoring whole-body fat using ORO. Similarly surprising, although *pho-1* RNAi suppressed DIO, it increased LD overall signal in the intestine of worms fed HFrD (Fig 6A and 6E). However, *pho-1* RNAi decreased the overall LD signal in the hypodermis (Fig 6C and 6E). The data then suggest that *pho-1* contributes to DIO mainly through enlargement of LDs in the intestine and increasing the abundance of LDs in the hypodermis, but not through changes in muscle LDs. From a different perspective, the observation that knock-down of all 3 lean genes caused a significant reduction in LD signal in the hypodermis of worms fed HFrD (Fig 6C and 6E) suggests that changes in hypodermal lipid content substantially contribute to DIO in *C. elegans*. Also intriguing, all lean genes suppressed the enlarged body size of worms fed HFrD (Fig 5C and 5E); therefore, it is reasonable to hypothesize that changes in hypodermal fat stores sizably contribute to obesity and the enlarged body size observed in obese *C. elegans*.

## DIO suppressors restore health biomarkers and lifespan of HFrD-fed *C. elegans*

DIO reduces *C. elegans* healthspan and lifespan (Fig 4I, 4J, 4K and 4L). Therefore, we next assessed whether the suppressors of DIO would also suppress the shortening of lifespan caused by a HFrD. For the DIO-suppresor genes that do not cause a developmental delay (*pho-1* and *Y71H10B.1*), the RNAi treatment was started on L1 larvae. For the gene causing developmental delay (*let-767*), RNAi was started at the L4 stage. Survival over time was assessed in three independent biological replicates (summarized in S3 Table). Only lifespan reduction or extension with p<0.05 (Gehan-Breslow-Wilcoxon test) in all three independent replicates was considered significant. We found that RNAi against *pho-1*, *let-767*, and *Y71H10B.1* partially suppressed the short lifespan associated with DIO (Fig 7A, 7B and 7C and S3 Table). On the other hand, knockdown of the same genes did not alter *C. elegans*'s lifespan significantly when

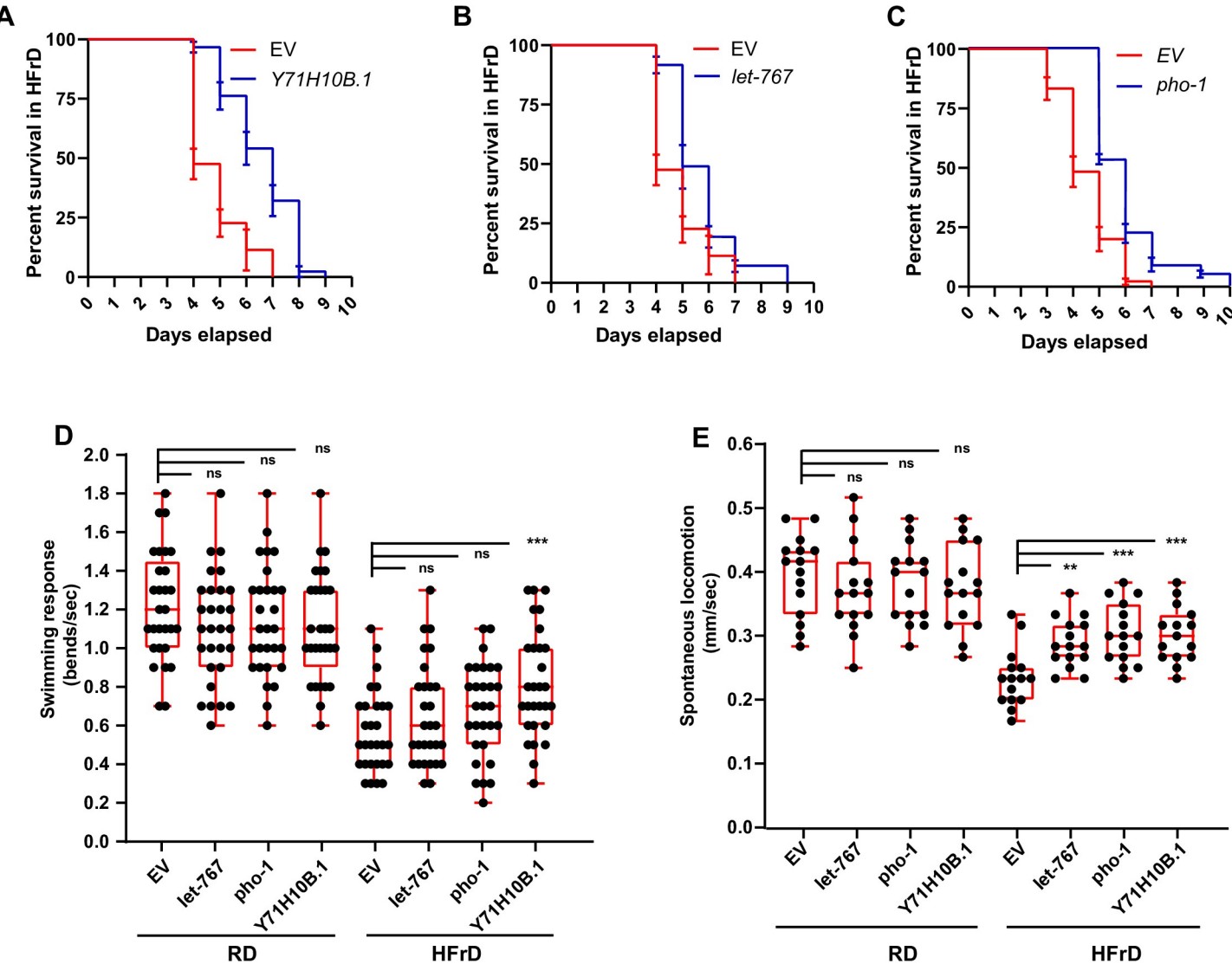

**Fig 7. The DIO suppressors ameliorate the negative effects of HFrD on *C. elegans* healthspan and lifespan.** Throughout this figure: Error bars = S.E.M. N = numbers of independent biological replicates. Unless specified, unpaired nonparametric t-tests were used to assess the significance. *p≤ 0.05, **p≤0.01, ***p≤0.001 and ****p≤0.0001. (**A**-**D**) Comparison of the lifespan of worms fed HFrD while treated with empty RNAi vector control (EV) or RNAi against the DIO suppressors: (**A**) *Y71H10B.1*, (**B**) *let-767*, and (**C**) *pho-1*. N = 1 is depicted in this figure. N = 3 is shown in S3 Table (**D**) Motor capacity of worms fed RD or HFrD and treated with RNAi against the DIO suppressors measured as bends/second after flooding as described in Materials and Methods section. N = 3. (**E**) Spontaneous displacement velocity on the surface of bacteria-free NGM plates of worms fed RD or HFrD and treated with RNAi against the DIO suppressors. N = 3.

animals are fed a regular diet (S3 Table). To examine if knockdown of the genes that reduce body fat also restores *C. elegans* health, we evaluated the behavioral response to flooding (swimming response) and spontaneous locomotion of animals with knockdown of the three DIO-suppressor genes (*let-767*, *Y71H10B.1*, and *pho-1*). We found that knockdown of *Y71H10B.1* ameliorated the swimming response (Fig 7D), and that knockdown of all of these genes ameliorated spontaneous locomotion in animals fed HFrD (Fig 7E). These findings support the notion that reducing body fat content can reduce the health and lifespan burdens associated with obesity in *C. elegans*, and provide evidence of the potential value of these genes as pharmaceutical targets.

## Discussion

GWAS is a powerful approach to identify loci associated with obesity because large cohorts are sampled across the arc of socioeconomic statuses, physical activity, eating habits, microbiota composition, and genetic diversity. However, connecting the resulting loci to the specific genes that promote or prevent obesity remains challenging. Obstacles include the large candidate-locus lists, defining the specific gene/s influenced by the genetic variants, and the time-consuming nature of molecular, cellular, and physiological studies [98]. For instance, a single decision such as where (*e.g.*, tissue) and when (*e.g.*, developmental stage) to inactivate or hyperactivate a candidate gene may change the outcome of the phenotypic assessment. As a result, comparatively few GWAS loci have been causally linked to the disease of interest. In the case of obesity, there are ~950 loci associated with BMI [51] and ~350 loci associated with waist-hip ratio adjusted BMI [52], yet very few genes have been validated and fewer druggable targets exist. Therefore, methods that can test the causality of GWAS loci quickly and effectively were needed.

In this study, we introduced two complementary approaches to validate potential obesity GWAS genes in a high-throughput manner: bioinformatic analyses of publicly available datasets and high throughput *in vivo* RNAi screening in the model organism *C. elegans*. This nematode has been used as a genetically-tractable animal model for uncovering and characterizing the cellular and molecular functions of genes related to complex human diseases such as obesity. Despite significant differences with mammals including lack of specialized adipose tissue for fat storage [99], absence of key mammalian fat regulators such as leptin [100], and a distinct cholesterol metabolism comparing to mammals [101], the core metabolic pathways (*e.g.*, Beta-oxidation) and signals regulating fat build-up and mobilization, fasting, healthspan and lifespan (*e.g.*, Insulin) are present and carry out comparable functions in both *C. elegans* and mammals [102]. Furthermore, key regulators of metabolism and lifespan were first discovered in *C. elegans* and then supported by mammalian studies (*e.g.*, DAF-16/FOXO [103]). Nevertheless, there are caveats associated with using RNAi in a nematode model system to test human obesity variants. The caveats range from false negatives due to the distinct biology and anatomy of *C. elegans* and mammals—as described above—to the fact that some human variants would be gains of function whereas RNAi causes loss of function. Additionally, gene-environment interactions are very complex and they play critical roles on the effect of gene variants. As such, the chosen genetic background may limit the discovery of genes with phenotypic effects only penetrant in specific mutant or wild-type genetic backgrounds. Nevertheless, as demonstrated in this study, combining gene candidate generation from human GWAS with testing causality via *in vivo* RNAi in *C. elegans* can aid in identifying genes contributing to complex human diseases such as obesity.

We here reveal 11 novel obese genes (*fbf-2*, *gon-1*, *hlh-2*, *let-355*, *mys-1*, *nst-1*, *pop-1*, *puf-8*, *rpac-19*, *rpt-5* and *Y46E12BL.2*) and two novel lean/DIO suppressor genes (*let-767*, and *pho-1*). We also retrieved three obese genes (*eif-6*, *glp-1* and *zfh-2*) and one lean/DIO suppressor gene (*Y71H10B.1*) that were previously linked to fat metabolism or obesity.

Among the known fat genes, we previously showed that inactivation of the *C. elegans* Notch receptor, *glp-1*, causes obesity [67]. By contrast, although the human orthologs of *glp-1*, *NOTCH4* and *EYS*, were associated with obesity in GWAS [56,57], they have not been causally linked to the disease. The combination of experimental validation of *glp-1* as an obesity gene in *C. elegans* with our database analyses showing that expression of *NOTCH4* and *EYS* negatively correlates with BMI and other markers of metabolic disease in mice and humans, puts the spotlight on *NOTCH4* and *EYS* as potential anti-obesity targets. However, not all hits from our screen showed a tight phenotypic correlation between worms and mammals. Unlike the

obesity phenotype observed upon whole-body knockdown of *eif-6* in *C. elegans* fed a regular diet, *Eif6* heterozygous mice had reduced body weight gain compared to their wild-type littermates [104]. Similarly, knockdown of *zfh-2* in *C. elegans* leads to obesity in animals fed a regular diet whereas heterozygous mutation of the mouse ortholog, *Zfhx3/Atbf1*, leads to reduced body weight gain [105]. Nevertheless, it is difficult to evaluate the body-weight phenotypes of *Eif6* and *Zfhx3* heterozygous mutant mice because these mutations lead to pleiotropic effects including perinatal mortality, growth retardation, and severe behavioral deficits [104,105], and neither body fat mass nor even circulating triglycerides were reported in the single *in vivo* mouse mutant studies available to date for each of these genes. Further emphasizing the need for future analyses of these genes, reduced expression of *Zfhx3* is strongly associated with increased body fat, fat mass, and markers of metabolic syndrome including insulin sensitivity and high cholesterol in mice (Fig 3A), suggestive of a role in obesity for this gene. Therefore, the apparent contradictory effects of inactivating *eif-6* and *zfh-2* in *C. elegans* and constitutive inactivation of the murine orthologs need further investigation.

In addition to the three previously characterized genes, we identified 14 *C. elegans* genes not yet causally linked to obesity. The biological functions of most of these genes and their mammalian orthologs have not been fully elucidated. Nonetheless, some have been characterized to a level that enables us to propose hypotheses about their roles in fat storage. For instance, *KAT8* –the human ortholog of *C. elegans mys-1* –promotes acetylation of fatty acid synthase (FASN), which leads to reduced FASN activity [106]. FASN is the terminal enzyme in *de novo* lipogenesis; therefore, it is reasonable to hypothesize that reduced KAT8 activity might increase lipogenesis and hence promote obesity, a phenotype that would be in line with the obesity phenotype we describe here for knockdown of the *C. elegans* ortholog *mys-1* and the negative correlation between *KAT-8* expression and obesity in humans. Another example, *TCF7L2* –the human ortholog of *C. elegans pop-1*– has been shown as a key regulator of glucose homeostasis. It has been reported that overexpression of a nuclear isoform of *Tcf7l2* (mice ortholog) in high-fat diet fed mice improves glucose tolerance, while depletion of *Tcf7l2* in mice causes higher glucose levels and impaired glucose tolerance [107,108]. Although the mechanism is not fully elucidated, impaired glucose homeostasis has been strongly associated with obesity in numerous studies [109]. The protective role of *Tcf7l2* in glucose homeostasis may suggest a protective role in obesity too. In support of this notion, expression of *Tcf7l2* in mice and *TCF7L2* in humans negatively correlates with BMI, which is in line with our observation that knockdown of *pop-1* promotes obesity in *C. elegans*. Therefore, future study of the role of *TCF7L2* in human obesity is warranted.

Increased sugar intake, especially fructose from high-fructose corn syrup, was suggested to be a leading cause of the current prevalence of obesity [12–14]. Fructose has been reported to have even more detrimental effects than glucose in healthspan [15]. Here we demonstrated that a HFrD not only disrupts lipid homeostasis but causes premature health deterioration a death in *C. elegans*; a result that is in line with two studies showing that dietary supplementation of 5 and 10% fructose reduces lifespan in *C. elegans* [48,49]. Interestingly, one of these studies also showed that feeding 1 or 2% fructose extended *C. elegans* lifespan [49]. This is intriguing because 1% is the dose of fructose we use in the present study, and we observed a dramatic reduction in the lifespan of *C. elegans* subjected to this dietary regimen. However, there are experimental details that may explain the discrepancies. Zheng *et al* supplemented fructose to worm media plates seeded with *E. coli* OP50, while we used plates seeded with *E. coli* HT115. We made the choice of performing our whole study on animals fed the RNAi-competent strain *E. coli* HT115 to maintain the dietary composition consistent between the primary screen and the follow-up studies. The relevance of *E. coli* to *C. elegans* metabolic phenotypes is well documented. Fat storage, healthspan, and lifespan depend on the bacteria

composition [84,110,111]. Further, recent large-scale *E. coli* mutant screens revealed microbial biochemical pathways influencing *C. elegans* fat accumulation [112] and longevity [113]. Based on these studies, we hypothesize that the discrepancies in the *C. elegans* lifespan phenotypes even when fed the same dose of fructose may be due to the use of different *E. coli* strains in the two studies. This is a very interesting observation since little is known about the molecular underpinnings defining the outcomes of different diet-microbiome-host three-way interactions in the development of obesity. Therefore, our *C. elegans* HFrD model can serve as an informative platform to molecularly dissect such complex interactions in future studies.

Importantly, in this study, we define that knockdown of *Y71H10B.1*, *let-767*, and *pho-1* prevents the development of DIO in *C. elegans*. The mammalian ortholog of *C. elegans* *Y71H10B.1*, named *Nt5c2*, encodes for a purine nucleotidase. *Nt5c2* knockout mice gain less weight when fed a high-fat diet [114], in line with the DIO suppressor phenotype observed in *C. elegans* treated with RNAi against *Y71H10B.1*. Furthermore, using adipose tissue gene expression from the METSIM cohort, we defined that the expression of the human ortholog, *NT5C2*, positively correlates with waist-to-hip ratio and circulating triglycerides, suggesting that inactivation of *Y71H10B.1/NT5C2.* would have an antiobesity effect in animal species ranging from *C. elegans* to humans. Similarly, mice carrying a mutation in *Acp2*, the murine ortholog of the *C. elegans* DIO suppressor gene *pho-1*, are smaller and gain less weight than the wild-type controls [115]. Further, again using the METSIM cohort, we found that the expression of the human ortholog, *ACP2*, positively correlates with body weight and total triglycerides. Interestingly, the function of *pho-1*-like genes in fat metabolism may be conserved even across kingdoms, as reduction of *ACP4* –the predominant ortholog of *pho-1* in *Arabidopsis*–leads to a decrease in total leaf lipids [116]. The congruency of phenotypic effects between inactivation of *Y71H10B.1* and *pho-1* in *C. elegans* and their orthologs in mammals supports the notion that these genes would be causally linked to obesity across animals. In addition the similarities strengthen our confidence in the value of our nematode screening approach to defining which GWAS loci are causally linked to obesity.

Given the causal link between obesity and age-related diseases [117], we sought to define whether knocking down of the three DIO suppressor genes–*pho-1*, *let-767*, and *Y71H10B.1*– ameliorates the negative impact of a high-fructose diet on *C. elegans* health and lifespan. We found all 3 DIO suppressors of obesity also partially suppressed the detrimental effects of HFrD on health and lifespan. To approximate whether the orthologs of *pho-1*, *let-767*, and *Y71H10B.1* may also impact healthspan in humans, we mined the GWAS catalog. One variant (rs75393320-C) that mapped to the human ortholog of *pho-1*, *ACP2*, is associated with HDL-C [118]. HDL-C has been suggested to have a protective role in cardiovascular health [119], which is a leading cause of premature death in humans.

With respect to *let-767*, its mammalian ortholog *Hsd17b12* is a member of the hydroxysteroid dehydrogenase superfamily specifically involved in the synthesis of arachidonic acid [120]. Knock out of *Hsd17b12* in mice causes embryonic death at the E8.5 stage [121], which is in line with the *C. elegans* developmental delay phenotype we report here. As for humans, we show that the expression of *HSD17B12* is positively associated with LDL-C and negatively associated with HDL-C, implying a detrimental effect of *HSD17B12* on health. Further, other 18 variants mapped to *HSD17B12* are associated with diseases including Type-2 diabetes, cardiovascular disease, and coronary artery diseases in the GWAS Catalog, suggesting a role for *HSD17B12* in the modulation of healthspan that is in line with our observation that depletion of *let-767* improves healthspan in *C. elegans*.

Regarding the human ortholog of *Y71H10B.1*, *NT5C2*, there are 77 variants mapped to *NT5C2* associated with a variety of disease-related traits (besides obesity traits), including hypertension, abnormal blood pressure, coronary artery disease, and cardiovascular diseases

in the GWAS Catalog. Moreover, knockout of *Nt5c2* (in mice) not only protects against high-fat diet-induced weight gain and adiposity but also improves insulin sensitivity and reduces hyperglycemia [114], all in line with our observation of improved healthspan and lifespan in *C. elegans* treated with RNAi against *Y71H10B.1*.

In summary, we identified 17 genes that promote or prevent *C. elegans* obesity, and the early onset of organismal deterioration and death associated with obesity. By combining the results of in *vivo* studies in *C. elegans* with analyses of mouse and human GWAS databases, we defined that the sign of the correlation between the expression levels of the mouse and human genes and their associated clinical traits matches, for the most part, the phenotypic effects of knocking down these genes in *C. elegans*, suggesting conserved causality and pharmacological potential for these obesity genes.

## Materials and methods

### *C. elegans* strains

*C. elegans* strains N2 (Bristol, UK), NL2099 *(rrf-3(pk1426))*, GRU101 (*gnaIs1 [myo-2p::yfp]*) and GRU102 (*gnaIs2 [myo-2p::YFP+unc-119p::Abeta1-42]*) were obtained from the Caenorhabditis Genetics Center. The tissue-specific lipid droplet reporter strains XD3971 (*xdIs143 [P$_{daf-22}$ PLIN1::GFP rol-6(su1006)]*), XD2458 (*xdIs56[P$_{Y37A1B.5}$ PLIN1::GFP rol-6(su1006)]*), and XD1875 (*xdIs26[P$_{unc-54}$ PLIN1::GFP rol-6(su1006)]*) were kind gifts from Dr. Monica Driscoll and Dr. Xun Huang.

### *C. elegans* media

NGM (nematode growth medium) agar plates were prepared as previously described [122]. RNAi NGM plates were prepared according to the same protocol, except that 1mM IPTG and 50μg/mL carbenicillin (final concentrations) were added to the molten agar. For 96-well RNAi plates, 200μL of the RNAi NGM was added to each well. For high-fructose diet (HFrD) plates, fructose was supplemented to the plates to reach a final concentration of 10mg/mL.

### *E. coli* culturing

For every biological replicate, fresh *E. coli* streaks or library stamps on LB-carbenicillin 50μg/mL (for RNAi clones) were used. Bacterial cultures were started from single colonies using a sterilized inoculating tip or hedgehog, and grown overnight for <16h in LB carbenicillin 50μg/mL in the absence of IPTG (or any other additives). For aeration, flasks were shaken at 250 rpm, and 1.2mL deep 96-well plates at 1,000 rpm. For targeted experiments, bacteria were harvested by centrifugation at room temperature and resuspended to OD600nm = 20 in S-buffer (~10X concentrated). For screening, 1.2mL bacterial cultures were resuspended with 20μl of S-buffer. Concentrated *E. coli* were seeded onto NGM or NGM-RNAi plates immediately and never exposed to the cold.

### RNAi screen

The RNAi screen for fat regulators was performed following our published protocol[39] with minor modifications briefly described here. The RNAi clones were inoculated in deep 96-well plates containing 1.2mL per well of terrific broth with carbenicillin 50μg/mL, and grown for 12-16h at 37˚C and 1,000 rpm. 50μl of 24x concentrated *E. coli* were seeded onto each well containing 200μl of NGM RNAi media. Seventy-five synchronized NL2099 (*rrf-3(pk1426)*) hatchlings were seeded per well and incubated for 60h at 15˚C and then 24h at 25˚C. Once the worms reached the young adult stage, they were harvested with S-buffer and stained with Oil

Red O (ORO) 0.5%, and imaged as previously described [39]. Individual worms in the microphotographs were identified and >500 parameters extracted using CellProfiler pipelines (cellprofiler.org/wormtoolbox) optimized as previously published [50,123]. The RNAi treatments were classified according to their fat phenotypes using the supervised-machine tool CellProfiler Analyst as previously described [50]. Phenotypic classes in the regular diet and the high-fructose diet screens were defined as described in the Results section.

The regular diet and the HFrD screens were independently repeated three times, and in each of these three experiments, each candidate obesity gene was inactivated twice (experimental replicates). RNAi treatments that consistently showed the same phenotype across experimental and biological replicates were selected for validation. For validation, single colonies of RNAi bacteria were inoculated into 500mL LB and incubated at 37°C, 200 rpm, overnight. Bacteria were then harvested by centrifugation at room temperature and resuspended to $OD_{600nm}$ = 20 in S-buffer (~10X concentrated) and seeded onto 6cm RNAi plates previously seeded with water (mock) or fructose 10mg/mL final concentration. Three hundred synchronized NL2099 hatchlings were seeded onto the plates and incubated at 15°C for 60h, followed by 24h incubation at 25°C. Worms were then harvested using S-buffer into 1.5mL tubes and stained with freshly made Oil Red O (ORO) and imaged as previously described [39,50]. ORO intensity was quantified using ImageJ (fiji.sc).

## Aging assay

*C. elegans* were reared on 6cm NGM or NGM RNAi plates seeded with 200μl of 20x concentrated suspensions of fresh overnight cultures of *E. coli* HT115 carrying empty vector RNAi control (L4440 plasmid) or RNAi plasmids targeting the genes of interest. The F1 of well-fed and uncontaminated gravid adults was synchronized using the hypochlorite/NaOH method. Once seeded onto plates, hatchlings were incubated at 15°C for 60h. Once they reached the L4-stage, 30 worms were transferred to two regular diet NGM RNAi plates and two high-fructose diet NGM RNAi plates. The plates were incubated at 25°C for the rest of the experiment. Starting twenty-four hours later (when worms were gravid adults) and until all animals in each population were dead, viability was scored daily by prodding. Worms were classified as dead and removed from the plates if they did not respond for 10 sec to gentle touch with a sterilized platinum wire. Worms were classified as "censored" if they were missing, displayed vulval rupture, or internal hatching. Data were analyzed using the Mantel-Haenszel method to calculate hazard ratio and 95% Cl, and the Gehan-Breslow-Wilcoxon method to calculate the p-value. The experiments were repeated a minimum of 3 independent times for each treatment tested.

## Healthspan assays

Healthspan assays were modified from previous publications [90,124]. 3d old adults were harvested from plates into 1.5mL tubes and washed twice with S-buffer. For the body bending assay, 30 3d old worms were added to one well of a 12-well plate (CytoOne) containing 2mL of S-buffer. The plates were kept at RT for 15min prior to imaging. A time-lapse (30s) was recorded using a Zeiss Axio Zoom.v16 dissecting microscope, PlanNeoFluar Z 2.3X/0.57 FWD objective, zoom 7X. The movie was analyzed to count body bends using the ImageJ plug-in wrMTrck(www.phage.dk/plugins/wrmtrck.html). For velocity measurements, 3d-old worms were harvested, washed, resuspended in S buffer, and the concentration of the suspension was estimated. Approximately thirty 3d-old worms were seeded onto an empty 6cm NGM plate (no bacteria). Once the drop of Sbuffer dried, we waited for 5 min at room temperature before starting imaging. Time-lapses (2min at 5fps) were captured using a Zeiss Axio Zoom.v16 dissecting microscope equipped with a PlanNeoFluar Z 2.3X/0.57 FWD objective, at 7X

magnification. Worm displacement velocity was scored using the ImageJ Manual Tracking plugin. For the racing assay, similar to the method described previously [97], 3d-old worms were harvested, washed, resuspended in S buffer, and the concentration of the suspension was estimated. Approximately fifty worms were then transferred to a 10cm NGM plate previously seeded with 25μL of 20X concentrated *E. coli* HT115 placed at the opposite side of the plate from where the worms would be seeded. After 1h, images of the mini lawns were taken using a Zeiss Axio Zoom.v16 dissecting microscope, PlanNeoFluar Z 2.3X/0.57 FWD objective, at 7X magnification. Using the images, the number of worms on the bacterial lawn or touching the lawn were manually counted.

## Lipid droplet live imaging and analysis

Day-1 adult worms carrying the lipid-droplet fluorescent reporters (XD1875, XD3971, or XD2458) were harvested, washed twice with S-buffer, paralyzed with 40mg/mL levamisole, and mounted onto agar pads as previously described [125]. Images were captured with a Nikon Eclipse Ti spinning disc confocal microscope, 60X/0.85NA objective, 500ms exposure time, and 80% laser intensity. The images were then analyzed using ImageJ to measure the intensity and the size of the lipid droplets. To measure the intensity, a ROI was defined and the measure function was used to measure the mean intensity per $\mu m^2$. To measure the LD diameter, random LD was picked and the diameter was measured by using the line and measure function.

## Statistics

All statistical analyses were performed using Graphpad Prism. Outliers were detected and removed from analyses using the ROUT method. For absolute intensity measurements, lipid droplet diameter, body size, and velocity, the unpaired nonparametric t-test was used to make single comparisons between each treatment and the mock control. The ratio t-test was used to compare all ratios. Aging statistics performed as described above. Unless otherwise stated, significance is represented as follows: $^*p \leq 0.05$, $^{**}p \leq 0.01$, $^{***}p \leq 0.001$, and $^{****}p \leq 0.0001$. Unless otherwise stated, data in this study are presented as mean values +/- SEM. All experiments were performed and quantitated at least three independent times.

## Human genetics database

Human gene expression data, clinical trait data, and SNP association data were previously collected and published by the Metabolic Syndrome in Men (METSIM) cohort [56]. This cohort has 770 healthy male subjects with mean BMI = 26.59 ± 3.47. Gene expression from subcutaneous adipose tissue was measured using Affymetrix U219 microarrays, and extensive clinical phenotyping was performed, including anthropometric traits, insulin sensitivity metrics, and blood metabolite levels. BMI and WHR were both reported. Correlations between gene expression and clinical traits were calculated using the biweight midcorrelation, and only significant associations are shown. Associations between SNPs and gene expression were calculated using a linear mixed model as described in Civelek et al [56]. The SNP with the most significant gene expression within 500 kB of the transcriptional start site was designated as the eSNP. We obtained the associations between BMI and WHR from Pulit et al [52]. LocusZoom plots were generated using the BMI and WHRadjBMI summary statistics from Pulit et al. and genotype and gene expression data from the METSIM cohort [56]. Plots were generated using the LocusZoom browser tool.

## Supporting information

**S1 Fig.** Throughout this figure: Scale bars = 200μm, Error bars = S.E.M. N = number of independent biological replicates. Statistical significance was assessed via ratio t-test, *p≤ 0.05, **p≤0.01, ***p≤0.001 and ****p≤0.0001. (**A**) Overview of the workflow of the screen for genes altering fat storage in *C. elegans*. (**B**) Representative images of the body fat content and distribution in WT(N2) and *rrf-3* RNAi sensitive mutant worms. N = >5. (**C**) Representative images of the body fat content and distribution as made evident with ORO in worms treated with RNAi against the obesity candidate genes from the L1 stage. Asterisks denote those RNAi treatments that led to developmental delay. (**D**) Representative images of the age-dependent increases in fat stores as revealed by staining L3, L4, and 1-day adult worms with ORO. N = >5. (**E**) Retesting of ORO staining in worms treated with *hsp-4* RNAi from the L3 stage showed no effect of *hsp-4* knockdown on *C. elegans* body fat content. (**F**) ORO quantification in worms treated with *hsp-4* RNAi as represented in panel **E**. Each data point represents the ORO intensity in one worm. Individual intensity values were normalized to the mean value of the EV RNAi control. N = 3.
(PDF)

**S2 Fig. Analysis of human orthologs association to BMI. A**) Association to BMI in TWINsUK **B**) Correlation with BMI in GTEx. Blue indicates the direction of association aligns with the lean gene in *C. elegans*. Red indicates the direction of association aligns with the obese gene in *C. elegans*.
(PDF)

**S3 Fig.** Throughout this figure: Error bars = S.E.M. N = numbers of independent biological replicates. Unless specified, unpaired nonparametric t-tests were used to assess the significance. *p≤ 0.05, **p≤0.01, ***p≤0.001 and ****p≤0.0001. (**A**) Quantification of muscle LD size in worms fed RD or HFrD. Each data point represents the measurement of random LDs in the images. ≥10 worms were measured in each independent biological replicate. N = 3. (**B**) Quantification of hypodermis LD size in worms fed RD or HFrD. Each data point represents the measurement of a random LD in the images. ≥10 worms were measured in each independent biological replicate. N = 3. (**C**) Body fat content of worms treated with RNAi against all of the DIO suppressors observed in the primary screen. RNAi was initiated from the L1 stage. *pho-1* and *Y71H10B.1* (Blue font) are the 2 DIO suppressors that cause leanness without developmental delay. (**D**) Representative images of the body fat content and distribution in worms fed HFrD from the L1 stage but treated with RNAi against *rpt-5* and *hsp-4* from the L4 stage. No change in fat content was observed in this condition. N = 3. (**E**) Representative images of the body fat content and distribution in worms treated with RNAi against *pho-1* from the L4 stage. No change in fat content was observed. N = 3
(PDF)

**S4 Fig.** LocusZoom plots of the associations of the SNPs near *ACP2* with (**A**) BMI and (**B**) *ACP2* gene expression in subcutaneous adipose tissue in the METSIM cohort. The effect of the alleles of rs10501321 on (**C**) *ACP2* expression and (**D**) total triglycerides. (**E**) Correlation between BMI and *ACP2* (human *pho-1* ortholog) expression in the METSIM cohort.
(PDF)

**S1 Table. Tabulated summary of the results of the GWAS meta-analysis and RNAi screen.** List of the 340 human genes associated with obesity traits extracted from 3 previous publications: 1) Mete Civelek, et. al, Am J Hum Genet., 2017 [56]; 2) Masato Akiyama, et.al, Nature Genetics, 2017 [57]; and 3) Audrey Chu, et.al, Nature Genetics, 2017 [58]. 207 of these human

genes had C. elegans orthologs. Since there is often more than one *C. elegans* ortholog per human gene, 386 *C. elegans* orthologs were actually identified to correspond to the 207 human genes. However, only 293 *C. elegans* genes were screened because there weren't available RNAi clones for 96 of the C, elegans orthologs. 14 obese, 2 generic lean, and 3 DIO-suppressor genes were identified and validated.
(XLSX)

**S2 Table. Lifespan analysis of worms fed HFrD versus RD.** Summary of 3 repeats of the lifespan analysis of worms fed HFrD versus RD. Hazard ratios were calculated using Log-rank and Mantel-Haenszel methods.
(XLSX)

**S3 Table. DIO suppressors extend lifespan of C. elegans fed HFrD.** Summary of 3 repeats of the lifespan analysis of worms treated with RNAi constructs versus EV in RD and HFrD conditions. Hazard ratios were calculated using Log-rank and Mantel-Haenszel methods.
(XLSX)

**S1 LocusPlots. LocusZoom plots of the associations of the variants within 1Mb of the human genes with BMI.**
(DOCX)

**S2 LocusPlots. LocusZoom plots of the associations of the variants within 1Mb of the human genes with WHRadjBMI in genome wide association studies.**
(DOCX)

**S1 Source Data.** This source data file includes all raw data and analysis presented in Figs 1E, 2B, 2C, 2F, 3A, 3B, 3C, 4B, 4C, 4E, 4F, 4G, 4H, 4J, 4K, 4L, 5B, 5C, 5D, 5E, 6B, 6C, 6D, 7D, 7E, S1E, S3A, S3B, S4A, S4B, S4C, S4D and S4E.
(XLSX)

**S2 Source Data.** This source data file includes all raw data and analysis of aging experiments, presented in Figs 4I, 7A, 7B, 7C, and S2 and S3 tables.
(XLSX)

## Acknowledgments

We thank Dr. Monica Driscoll and Dr. Xun Huang for generously sharing the LD reporter strains. We especially thank Nella Solodukhina for helping manage the lab, preparing reagents, and conducting some of the experiments. We acknowledge the 2020 and 2021 BIOL4005 class of the University of Virginia for their help in cross-validating automated scoring.

## Author Contributions

**Conceptualization:** Wenfan Ke, Eyleen J. O'Rourke.

**Data curation:** Wenfan Ke, Jordan N. Reed, Chenyu Yang, Noel Higgason, Leila Rayyan, Mete Civelek, Eyleen J. O'Rourke.

**Formal analysis:** Wenfan Ke, Jordan N. Reed, Chenyu Yang, Noel Higgason, Leila Rayyan, Carolina Wählby, Anne E. Carpenter, Mete Civelek, Eyleen J. O'Rourke.

**Funding acquisition:** Carolina Wählby, Anne E. Carpenter, Mete Civelek, Eyleen J. O'Rourke.

**Investigation:** Wenfan Ke, Jordan N. Reed, Chenyu Yang, Noel Higgason, Leila Rayyan, Carolina Wählby, Anne E. Carpenter, Mete Civelek, Eyleen J. O'Rourke.

**Methodology:** Wenfan Ke, Jordan N. Reed, Carolina Wählby, Anne E. Carpenter, Eyleen J. O'Rourke.

**Project administration:** Eyleen J. O'Rourke.

**Resources:** Wenfan Ke, Eyleen J. O'Rourke.

**Software:** Wenfan Ke, Jordan N. Reed, Carolina Wählby, Anne E. Carpenter, Mete Civelek, Eyleen J. O'Rourke.

**Supervision:** Mete Civelek, Eyleen J. O'Rourke.

**Validation:** Wenfan Ke, Chenyu Yang, Noel Higgason, Leila Rayyan, Eyleen J. O'Rourke.

**Visualization:** Wenfan Ke, Jordan N. Reed, Mete Civelek, Eyleen J. O'Rourke.

**Writing – original draft:** Wenfan Ke, Jordan N. Reed, Eyleen J. O'Rourke.

**Writing – review & editing:** Wenfan Ke, Jordan N. Reed, Anne E. Carpenter, Mete Civelek, Eyleen J. O'Rourke.

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
