## [Decision Letter · Decision Letter 0]

2 Jul 2021

Dear Dr O'Rourke,

Thank you very much for submitting your Research Article entitled 'Genes in human obesity loci are casual obesity genes in C. elegans' to PLOS Genetics. The manuscript was fully evaluated at the editorial level and by three independent peer reviewers. The reviewers were quite enthusiastic about the paper, but also identified some minor concerns that we ask you address in a revised manuscript.

We therefore ask you to modify the manuscript according to the review recommendations. Your revisions should address the specific points made by each reviewer.

[LINK]

Yours sincerely,

Gregory P. Copenhaver

Editor-in-Chief

PLOS Genetics

Reviewer's Responses to Questions

**Comments to the Authors:**

Reviewer #1: In this manuscript, the authors took on the Herculean task of validated human GWAS hits for obesity and related disorders (BMI, etc.). It has long been established that these traits have a strong genetic component, but very few genes have been validated in human or mammalian model systems. Using fat staining, high-throughput assays, diet-induced obesity models, and RNAi, the authors test 293 genes for their roles in C. elegans obesity. They make compelling cases for 14 genes that inhibit obesity and three genes that promote obesity. This second class offers potential druggable targets to reduce obesity in humans. The manuscript reads well and the experiments are well done. I only have two minor criticisms that should be addressed.

Minor criticisms:

1. RNAi of the gene rpt-5 caused different fat defects depending on when the gene was inhibited during C. elegans development. The authors decided to classify this gene with 13 post-developmentally obese genes. However, the authors did not specify whether the plate-based RNAi, where 13 of 14 genes retested the 96-well assay phenotype, was performed on L4 animals. It did not seem that way in the text. So, the authors should specify that the RNAi for these 14 genes (not rpt-5) was post-L4 or be more specific in their classification.

2. High-fructose diet induces behavior effects similar to A-beta over expression. However, that similarity does not mean that they act in the same pathway. The result is a mere correlation. The authors should tone down their conclusions from lines 394-404 and in parts of the Discussion.

Specific edits:

Line 98, space between association and from

Line 103 and throughout, e.g. should be italicized and have a comma

Throughout, numbers less than 10 should be written and not numeric

Line 193 (and throughout), every organism has a phenotype. It is either wild-type or mutant. I think the authors mean mutant phenotype here

Line 206, please provide references

Line 243, extra space after 16

Reviewer #2: Ke et al, PLoS Genetics

In this fascinating paper, authors analyzed human GWAS meta-analysis studies focused on obesity associated traits—to identify 386 candidate orthologs in C. elegans; authors tested 293 (RNAi knockdown reagents readily available) for elevated fat as monitored by Oil O red staining. Studies identified 14 genes for which kd increased fat, and two that decreased fat in post-development animals. The authors cataloged how expression “direction” data on most human homologs correlate with measures of nematode obesity (i.e. homolog also down regulated in obese state, and/or metabolic disease. A second facet of the work addresses impact of growth on high fructose, a condition in which body fat increases, and lifespan, locomotory healthspan, and resistance to Abeta toxicity in neurons decreases; impact on lipid droplet accumulation is also assayed. The authors proposed high fructose as a high fat diet model and identify homologs of 4 human GWAS genes that can modulate high fat increase induced by fructose; one of which, pho-1, appears specific to the high fructose diet, focused in hypodermis.

Data contribute novel hypothesis for action of some human GWAS identified genes, suggesting priorities for extended studies by showing causality of outcome in the worm model.

Studies are well documented, with deep phenotypic outcomes rigorously demonstrated for multiple measures. Premise and outcome are quite exciting in the field, as they highlight a set of genes that modulate C elegans baseline fat content, and/or diet-regulated fat contents, they underscore potential importance of targeting human homologs for modulation of obesity. I recommend publication on the basis of these strong points: 1) the premise, that genetic perturbation of GWAS-highlighted candidates for obesity modulators in C. elegans will identify conserved modulators of fat content is strong; and the hypothesis is borne out to a large extent, at least for initial analysis. The basic concept has precedent in the field, and outcome here provides a particularly strong example of the value of this approach. 2) Outcomes identify novel targets for focused study in worms and in mammals regarding fat management and health.

Points to consider for revision:

5% fructose impact on longevity and mobility healthspan has been in a 2020 paper that focused on mitochondrial outcomes (PubMed 33258406). Roles as a high fat model are novel to the work here and merit original publication; documentation is of higher rigor than existing literature, and extends phenotyping in important ways. However, the authors should cite this work, and briefly summarize findings. The authors might convert the 5% value to compare to the 10mg/ml concentration used in this study in comments.

Likewise, PMID 27680107 reports on low dose fructose for lifespan extension; it is of interest to the field for He et al to cite this and to point these data out, with comment if possible.

The authors document that use of an rrf-3 mutant to potentiate RNAi does not change baseline fat phenotypes, an important control. Still, the targeted screen in this background might miss critical genes expressed in neurons and rrf-3 deficits could change the interactions with the genes identified. Minimally, the authors should discuss this caveat relative to outcome. It would be informative to test in wt background, and someday (not for this paper) retest the candidate set in a more directed neuronally-sensitized strain.

The senior author is well known for her contributions to the use of Oli red O for analysis of C. elegans fat stores, and her work highlighted that this reagent detects fat in intestine, hypodermis, germline and eggs. Thus, lack of germline/eggs, i.e. the sterile class of animals, might be anticipated to have low staining—that 9 genes for which kd enhances fat actually increase is therefore particularly striking, and suggests a possible change in fat tissue distribution when germline/eggs are lacking. The authors might note this mechanistic possibility when commenting on the obesity/infertility connection.

That puf-8 stands out as the exception to the joint disruption of fertility and fat accumulation is interesting; a bit more comment on that issue could be warranted.

Lines 297, 298 that human and elegans homologs are regulated in the same direction is interesting and worth pointing out. Still, correlations do not exactly speak to “conserved evolutionary function”—it is more precise to say conserved regulation.

Page 13 section regarding Figure 3C is a bit unclear. The legend refers to published GWAS studies, the narrative in the text refers to the not-yet-cataloged MRTSIM loci but the analysis relies on what are in the published GWAS —can the authors clarify the main point of this panel in the text and legend? I take it that 15/16 of genes identified have published GWAS data that links them to metabolic measures; what is the main point of the pleiotropic effects? This might be more clearly articulated.

Line 364, discussion around line 373, again 533—reference to the long lived daf-2 mutant having elevated fat stores. If high fat tends to be associated with short life, the daf-2 mutant shows the opposite. Some discussion of this paradox might be informative; at least the paradox might be recognized.

Line 407 wording is awkward. Better: We tested the 293 C. elegans orthologs of the human obesity GWAS candidate genes for DIO causality by screening…..

It is common to demand mechanism details in review; I think these data are adequately impactful on their own, setting the stage for further drill down in both worms and mammals

Reviewer #3: In this paper, the authors identify 293 genes from obesity GWAS studies in humans and perform analysis on these genes in a multiple c. elegans models, including a fructose feeding model the authors develop. In the original screen, the authors identified 23 genes that had an effect on lipid accumulation in c. elegans when the gene was silenced. The authors then develop a fructose fed model in c. elegans that promotes lipid accumulation. In this model they screened the 293 genes and found 8 genes with impacts on lipid accumulation. The authors wind down to 3 key genes that have significant effect on lipid accumulation and test these genes in another c. elegans model (Plin:GFP). Overall, the paper is well-written and presents an interesting high-throughput model to systematically screen genes identified from GWAS studies. I have a few comments below that would help improve the paper.

1. The identification of the 293 genes from obesity GWAS is quite brief in the paper. It would be helpful to the reader if the authors would provide a bit more detail into how they got to the 293 genes.

2. What is the logic of testing all the co-morbidities and healthspan parameters in the HFrd system? I had a bit of trouble following the logic in the paper for this.

3. As many of the people reading this paper would be interested in the human gene name, it would be helpful to see those gene names integrated into the text earlier on in the manuscript.

4. Could you show the human locus plots for the key genes you validate in your study?

5. Certainly there are limitations of using c. elegans in studying such a complex trait as obesity. It would be helpful to have a formal paragraph on the limitations of the model.

**Have all data underlying the figures and results presented in the manuscript been provided?**

Reviewer #1: Yes

Reviewer #2: Yes

Reviewer #3: Yes

PLOS authors have the option to publish the peer review history of their article (what does this mean?). If published, this will include your full peer review and any attached files.

Reviewer #1: No

Reviewer #2: No

Reviewer #3: No

---

## [Editor Report · Decision Letter 1]

23 Jul 2021

Dear Dr O'Rourke,

We are pleased to inform you that your manuscript entitled "Genes in human obesity loci are causal obesity genes in C. elegans" has been editorially accepted for publication in PLOS Genetics. Congratulations!

Yours sincerely,

Gregory P. Copenhaver

Editor-in-Chief

PLOS Genetics

Comments from the reviewers (if applicable):

**Data Deposition**

http://datadryad.org/submit?journalID=pgenetics&manu=PGENETICS-D-21-00734R1

**Press Queries**

---

## [Editor Report · Acceptance letter]

31 Aug 2021

PGENETICS-D-21-00734R1 

Genes in human obesity loci are causal obesity genes in C. elegans 

Dear Dr O'Rourke, 

We are pleased to inform you that your manuscript entitled "Genes in human obesity loci are causal obesity genes in C. elegans" has been formally accepted for publication in PLOS Genetics! Your manuscript is now with our production department and you will be notified of the publication date in due course.

With kind regards,

Andrea Szabo

PLOS Genetics

On behalf of:
